# Understanding rate effects in injection-induced earthquakes

Maryam Alghannam[1] & Ruben Juanes [1]✉

Understanding the physical mechanisms that underpin the link between fluid injection and seismicity is essential in efforts to mitigate the seismic risk associated with subsurface technologies. To that end, here we develop a poroelastic model of earthquake nucleation based on rate-and-state friction in the manner of spring–sliders, and analyze conditions for the emergence of stick-slip frictional instability—the mechanism for earthquakes—by carrying out a linear stability analysis and nonlinear simulations. We find that the likelihood of triggering earthquakes depends largely on the rate of increase in pore pressure rather than its magnitude. Consequently, fluid injection at constant rate acts in the direction of triggering seismic rupture at early times followed by aseismic creep at late times. Our model implies that, for the same cumulative volume of injected fluid, an abrupt high-rate injection protocol is likely to increase the seismic risk whereas a gradual step-up protocol is likely to decrease it.

---

[1] Massachusetts Institute of Technology, 77 Massachusetts Ave, Cambridge, MA 02139, USA. ✉email: juanes@mit.edu

Subsurface fluid-injection operations have been recognized to carry a risk of inducing earthquakes since the 1960s[1]. While most of the injection-induced earthquakes are micro-tremors, they can occasionally be of large magnitude, such as the Prague $M_w$ 5.7 earthquake in 2011[2], the Pawnee $M_w$ 5.8 earthquake in 2016[3], and the Pohang $M_w$ 5.5 earthquake in 2017[4], among others. The occurrence of induced earthquakes of large magnitude has motivated development of different operational strategies for seismic hazard mitigation. In particular, an early attempt to control seismicity at the Rangely oil field suggested maintaining the magnitude of fluid pressure below a critical threshold[5], based on a Coulomb failure model that links the magnitude of fluid pressure to the occurrence of induced earthquakes[6]. This model, however, does not address the evolution of the rupture and whether a fault slips seismically or aseismically. It was also insufficient to explain seismicity at Cogdell oil field, for instance, where earthquakes were observed in regions of low rather than high fluid pressure[7]. A different strategy to control seismicity involved maintaining the cumulative volume of injected fluid below a critical threshold[8], based on empirical observations and modeling linking the cumulative volume of injected fluid to the maximum magnitude of induced earthquakes[9]. This model, however, is at odds with the 2017 Pohang earthquake, as its magnitude exceeded the size estimated from the injected volume by 500 times[10].

A growing number of field observations suggests that managing fluid injection rates may be a promising tool to mitigate the occurrence of induced earthquakes. It is observed that low-rate wells, for instance, are much less likely to be associated with earthquakes than high-rate wells, and that the critical rate above which earthquakes are induced is likely dependent on reservoir properties[11,12]. It is also observed that temporal variation in injection rates is generally correlated with the frequency of earthquakes[1,13–15], and that abrupt increases in injection rates tend to shortly precede the occurrence of earthquakes[16–18]. While attempts have been made to explain some of these observations with seismicity-rate models[19–21] and 2D numerical simulations of coupled flow-geomechanics[22,23], the physical mechanisms behind the link between the rate of fluid injection and the occurrence of induced earthquakes remain poorly understood.

Seismicity-rate models based on the triggering-front concept consider the large-scale spatiotemporal effects of nonlinear diffusion on the probability of a given magnitude earthquake using Gutenberg–Richter statistics[24], but do not address the dynamics of the rupture and, in particular, whether a fault slips seismically or aseismically. Characterizing fault slip mode is essential to mitigate the seismic risk associated with subsurface operations, as it has been observed that an increase in pore pressure magnitude leads to seismic slip in some sites[1–5,25] and aseismic slip in others[7,26–30]. Here we develop a poroelastic model of induced earthquake nucleation in the manner of spring–sliders[31–35] based on rate-and-state friction[32,36], and we study the effect of injection rate on stick-slip frictional behavior—the mechanism for seismic slip[37].

Our model shows that the likelihood of triggering earthquakes depends critically on the rate of increase in pore pressure. We find that fluid injection at constant rate acts in the direction of triggering seismic rupture at early times followed by aseismic creep at late times. This finding is qualitatively consistent with laboratory observations of sliding between saturated rocks at both transient and steady-state pore pressure conditions[38–40], and may explain field observations of triggered and induced seismicity from subsurface operations in different geologic settings[1,11–14,16–18].

## Results and discussion

**Poroelastic spring–slider model.** When fluid is injected into a faulted reservoir, the pore pressure change induces effective stress

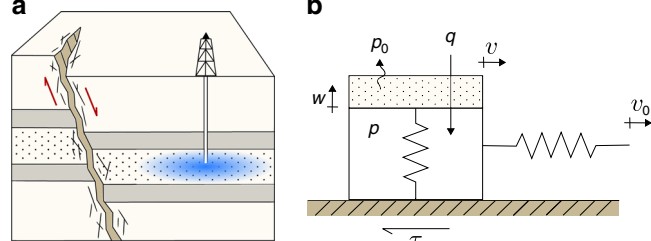

**Fig. 1 Conceptual picture of induced seismicity including poroelastic effects. a** Fluid injection induces effective stress changes in the reservoir surrounding the fault, increasing the likelihood of fault slip and earthquake triggering. **b** Our spring–poroslider model of a fault segment in contact with the reservoir. Here, $v_0$ is the loading velocity, $v$ is the velocity of the block, $\tau$ is the frictional shear force, $w$ is the elongation of the piston, $q$ is the injection rate, $p_0$ is the ambient pressure, and $p$ is the pressure inside the slider.

variations in the reservoir and surrounding rock (Fig. 1a). To model the effects of variations in effective normal stress on a creeping fault segment, we develop a poroelastic spring–slider model of frictional slip (Fig. 1b). Our model consists of a slider of unit base area that is pulled by a spring whose end is constrained to move at a steady slip rate. The spring stiffness accounts for the elastic interaction of the sliding surface with the surrounding rock. The slider represents the injection-driven deformation process, where a piston is loaded vertically and compresses a spring inside a fluid-filled space. The vertical spring is analogous to the rock skeleton, while the fluid inside the slider represents fluid in the rock pores subject to increase from fluid injection and decrease from pressure diffusion. Our model accounts for the poroelastic coupling between the shear and effective normal stresses along the fault.

Frictional evolution is modeled by the rate-and-state constitutive laws[32,36], which are capable of reproducing a wide range of observed seismic and aseismic fault behaviors ranging from preseismic slip and earthquake nucleation to coseismic rupture and earthquake afterslip[41]. These laws propose that the frictional shear stress is a function of the effective normal stress and a coefficient of friction that is dependent on slip rate and the state of the sliding surface. Since the effective normal stress varies as a result of fluid injection, we adopt Linker and Dieterich's[42] formulation for the coefficient of friction and couple it with a poroelastic model of pore pressure and rock deformation. We derive the poroelastic model from the principles of mass and momentum conservation, and find that pore pressure satisfies a diffusion equation that leads to transient behavior at early times and steady-state behavior at late times.

The dimensional equations describing the dynamic motion of the poroelastic spring–slider system with an evolving pore pressure take the form (see Supplementary Note 1 for the derivation of the equations):

$$\dot{U} = V_0 - V, \tag{1}$$

$$\dot{V} = \frac{1}{(T/2\pi)^2}\left[U - \frac{1}{k_s}\left(\mu_* + \hat{a}\ln\frac{V}{V_*} + \Theta\right)(\Sigma - P)\right], \tag{2}$$

$$\dot{\Theta} = -\frac{V}{d_c}\left(\Theta + \hat{b}\ln\frac{V}{V_*}\right) + \hat{\alpha}\frac{\dot{P}}{(\Sigma - P)}, \tag{3}$$

$$\dot{P} = \frac{k_n^{eff} k}{\eta L}(P_0 - P) + k_n^{eff} Q. \tag{4}$$

where $U$ is the relative displacement between the load point and the slider, $\dot{(\,)}$ denotes time derivative, $V_0$ is the loading velocity, $V$

is slip rate, $T$ is the vibration period, $k_s$ is the shear stiffness, $V_*$ is a normalizing slip rate, $\mu_*$ is a constant appropriate for steady-state at slip rate $V_*$, $\hat{a}$ and $\hat{b}$ are experimentally derived parameters relating friction to changes in slip rate and state, respectively, $\Theta$ is a state variable describing the sliding surface, $\Sigma$ is the total stress, $P$ is the pressure inside the slider (pore pressure), $d_c$ is the characteristic slip distance, $\hat{\alpha}$ is a scaling factor ranging from 0 to $\mu$[42,43], $k_n^{\text{eff}}$ is the effective normal stiffness (related to the uniaxial bulk modulus or the reciprocal of the uniaxial specific storage per diffusion length in a continuum), $k$ is the permeability, $\eta$ is fluid dynamic viscosity, $L$ is the pressure diffusion length, $P_0$ is the ambient pressure, and $Q$ is the volumetric injection rate per unit area.

Choosing the following characteristic quantities: $u_c = d_c$, $v_c = V_*$, $\mu_c = \mu_*$, $p_c = P_0$, $\tau_c = \mu_*(\Sigma - P_0)$, $\theta_c = \mu_*$, and $t_c = d_c/V_*$, the equations describing the dynamic motion of the system, in dimensionless form, become (see Supplementary Note 2):

$$\dot{u} = v_0 - v, \tag{5}$$

$$\dot{v} = \frac{1}{\epsilon^2}\left[u - \frac{1}{\kappa}(1 + a\ln v + \theta)(\sigma - p)\right], \tag{6}$$

$$\dot{\theta} = -v(\theta + b\ln v) + \alpha\frac{\dot{p}}{(\sigma - p)}, \tag{7}$$

$$\dot{p} = c(p_0 - p) + rq, \tag{8}$$

where $\kappa = (k_s d_c)/\tau_c$, $a = \hat{a}/\mu_c$, $b = \hat{b}/\mu_c$, $\alpha = \hat{\alpha}/\mu_c$, $\epsilon = (T/2\pi)/t_c$, $c = t_c/(\eta L/k_n^{\text{eff}}/k)$, $r = t_c k_n^{\text{eff}}$, and $q = Q/p_c$. The parameter $\kappa$ is the normalized shear stiffness, and $a$, $b$, $\alpha$ are normalized frictional parameters. The parameter $\epsilon$ is the normalized oscillation period or ratio of inertial to state-evolution timescales, which may range from $10^{-8}$ to $10^{-6}$ depending on rupture diameter and shear wave speed. The parameter $c$ is the normalized diffusivity or ratio of the pore pressure to the state-evolution timescales, which may range from $10^{-4}$ to $10^1$ depending on reservoir permeability, uniaxial bulk modulus, and well-fault distance. The parameter rq is the normalized injection rate, which may range from $10^{-5}$ to $10^{-1}$ depending on injection rate and reservoir size.

**Stability analysis.** The stability of steady frictional sliding to small perturbations in velocity, which determines whether motion is by slow steady-sliding or violent stick-slip, depends on the evolution of the frictional resistance. Stick-slip occurs whenever a change of frictional resistance with sliding occurs at a rate greater than the loading system is capable of following[31]. At a constant pore pressure, linear stability analysis of the system about steady-state leads to the stability condition by Ruina[32]. Pore pressure, however, is not constant in time and its evolution depends on the injection rate and on the poroelastic and hydraulic parameters of the rupture. To quantify this, we carry out a linear stability analysis of the system about a quasi steady-state where sliding is steady but pore pressure is evolving as a result of fluid injection. We find that motion is by stick-slip when the dimensionless shear stiffness of the loading system is lower than a critical value ($\kappa < \kappa_{\text{crit}}$) given by

$$\kappa_{\text{crit}} = (b - a)(\sigma - p) + \frac{\alpha}{v_0}\dot{p}, \tag{9}$$

and is by steady-sliding otherwise ($\kappa > \kappa_{\text{crit}}$). Variables $p$ and $\dot{p}$ are dimensionless pore pressure magnitude and pore pressure rate, respectively, at any point in time. Accordingly, frictional instability for the spring–poroslider system with an evolving pore pressure depends not only on the magnitude of pore pressure, but also on the rate of change of pore pressure (Supplementary Notes 3, 4, 5, and 6 for the analysis of QSSA, derivation of Eq. (9), and validation against nonlinear simulations).

As pore pressure evolves from initial to steady-state conditions in response to fluid injection, we find that the competing effects of $p$ and $\dot{p}$ exhibit a transition in their dominance over frictional instability (Fig. 2a). The destabilizing effect of $\dot{p}$ dominates when pore pressure grows rapidly at early times, resulting in an increase in critical stiffness (dashed curve in Fig. 2a). It then decreases as pore pressure diffuses and approaches steady state, giving rise to the stabilizing effect of $p$, which explains the decrease in critical stiffness at late times (dotted curve in Fig. 2a). This result is generally consistent with a linear stability analysis of slow slip with mildly rate-strengthening friction in a poroelastic con-tinuum[44], in which undrained slip-induced poroelastic pressure has a destabilizing effect and a sufficiently fast equilibration process has a stabilizing effect. In our analysis, the early-time destabilizing effect of $\dot{p}$ is likely attributed to a short-term effect

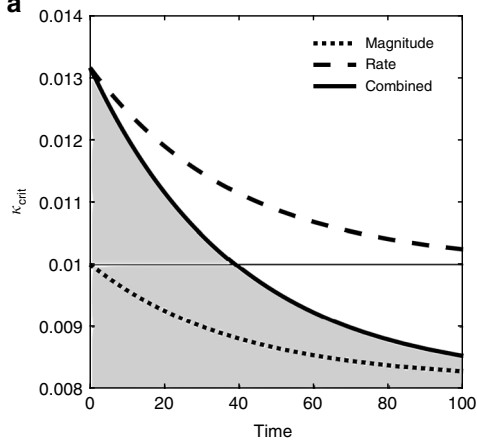

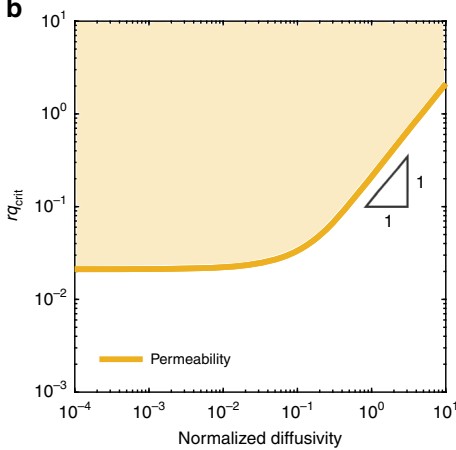

**Fig. 2 Dimensionless critical stiffness and critical injection rate. a** Critical stiffness—below which stick-slip instability is triggered ($\kappa < \kappa_{\text{crit}}$) and above which stick-slip instability is inhibited ($\kappa > \kappa_{\text{crit}}$)—in response to fluid injection at constant rate in velocity-weakening material ($c = 3 \times 10^{-2}$, rq $= 5 \times 10^{-3}$). The combined effect is illustrated by the solid curve, which shows a transition in the effect of fluid injection from de-stabilizing at early times to stabilizing at late times. The effect of the magnitude of pore pressure is illustrated by the dotted curve. The effect of the rate of change in pore pressure is illustrated by the dashed curve. **b** Critical injection rate—above which stick-slip instability is triggered—as a function of the normalized diffusivity of the system. The diffusivity is varied by varying permeability (beige curve in **b**).

on contact interlocking, where a decrease in effective normal stress results in fault opening and loss of asperity contacts[45]. The late-time stabilizing effect of $p$, in contrast, is likely attributed to a long-term effect on interface locking. A low effective normal stress tends to reduce the degree of interface locking, and thus limit the magnitude of stress drops[46–48].

This behavior is also qualitatively consistent with laboratory observations of sliding between saturated rocks at both transient and steady-state pore pressure conditions. The early-time destabilizing effect agrees with experimental studies showing that a gradual increase in pore pressure results in unstable slip during the transient-state, and that the degree of instability measured by the total slip, slip rate, and shear stress drop correlates with the rate of pore pressure increase[38,39]. The late-time stabilizing effect also agrees with another experimental study, showing that sliding between two rock surfaces is much more stable at high than at low steady-state pore pressure[40]. Sliding is observed to be by slow, steady-type motion at high pore pressure, and by stick-slip at low pore pressure. This behavior, however, is different from the observations of shearing granular fault gouge materials[49,50], where the frictional parameter $a-b$ is observed to decrease in magnitude with increasing steady-state pore pressure, an effect related to shear-induced dilatancy strengthening and pore compaction creep.

**Application to the Denver earthquakes.** Our findings, if they are applicable to natural faults, hold interesting and important implications for induced seismicity. The poroelastic spring–slider may be viewed as a simple model of a fault segment in contact with a reservoir, steady-sliding as an analog of aseismic creep, and stick-slip as a seismic wave-producing rupture cycle[33,37]. The spring stiffness scales inversely with the size of the fault segment[51]. Within this view, our findings may be generalized to indicate that a slowly creeping fault segment is destabilized and nucleates an earthquake if its size exceeds a critical value known as the nucleation length, which is inversely proportional to the critical stiffness in Eq. (9).

To bridge the gap between the analysis of the idealized poroelastic spring–slider model and the real world, we extend our instability criterion from dimensionless to dimensional form, and identify values of dimensionless parameters $c$ and rq that correspond to a real-world setting. The 1960s Denver earthquakes is a good example of a real-world setting, where it is well-documented that injection of wastewater into the fractured Precambrian granite gneiss underneath the Rocky Mountain Arsenal triggered the earthquakes and where injection rate is directly related to the frequency of earthquakes[52,53]. We find that reasonable estimates of $c$ and rq for this setting are in the order of $10^{-2}$ to $10^{-1}$ and $10^{-3}$ to $10^{-1}$, respectively. We then assess the effect of fluid pressurization by evaluating its contribution to the critical stiffness in Eq. (9). We find that a reasonable estimate for the increase in critical stiffness at early times is ~300%, which indicates that the weakening effect from fluid pressurization is likely significant in this setting (see Supplementary Note 7 for more details on the application to the Denver earthquakes).

When earthquakes nucleate on a fault with velocity-weakening friction, in general, aseismic creep may begin in sections of favorable stress conditions. The aseismically creeping segment then slowly grows in size until it reaches the nucleation length, and then it breaks out rapidly into a seismic wave-producing rupture[54]. A significant increase in critical stiffness, or equivalently decrease in nucleation length, from fluid pressurization may further facilitate or exacerbate this breakout—potentially increasing the likelihood of triggering earthquakes.

**Influence of reservoir properties on injection-induced seismicity.** To study the influence of reservoir properties on the critical injection rate, above which earthquakes are induced, we model the occurrence of earthquakes as a function of dimensionless injection rate $rq = d_c k_n^{eff} Q/(p_c V_*)$ and normalized diffusivity $c = d_c k_n^{eff} k/(\eta L V_*)$ for velocity-weakening conditions, $b - a > 0$. We find that the dimensionless critical injection rate is directly proportional to the diffusivity $c$ in the high-diffusivity limit, and independent of it in the low-diffusivity limit (Fig. 2b). These findings suggest that reservoir regions with low hydraulic diffusivity are more prone to induced seismicity than regions with high hydraulic diffusivity—a result that qualitatively agrees with the triggering-front concept[24] (see Supplementary Note 8 for more details on the phase diagram of injection-induced seismicity).

**Influence of injection strategy on induced seismicity.** The earthquake likelihood is strongly dependent on the duration of injection. For a fixed total injected volume, a shorter injection duration (or, equivalently, a higher injection rate) results in a higher likelihood of earthquake triggering (Supplementary Note 9).

To further understand how injection rate may be used to minimize or mitigate the seismic hazard, we simulate three different injection scenarios, and examine the stability of each. Figure 3 demonstrates how injecting the same volume of fluid can have very different seismic potential depending on the injection profile. We observe that injecting at constant rate in scenario (A) causes the critical stiffness to increase at early times, potentially triggering earthquakes, and decrease at late times, potentially resulting in the cessation of earthquakes. In addition, we observe a dramatic drop in critical stiffness upon stopping injection followed by recovery to the value prior to injection. Scenario (B) shows that a higher injection rate yields higher critical stiffness, implying an increased risk of seismicity for this higher injection rate. Scenario (C), where the injection rate ramps up in stages, seems to be most stable because the maximum critical stiffness is lower than its value in both (A) and (B). If the duration of each stage is not sufficiently long for pressure to stabilize at the fault, the ramp-up injection strategy does not counteract the destabilizing effect of the rate of pore pressure increase at each injection rate increment. In the Basel enhanced geothermal site, for example, the duration of injection stages was 1 day[55], while the time for pressure to stabilize at the fault is longer than 1 month—a conservative estimate based on the distance to the nearest fault segment to the injection well and the permeability of the fractured rock[25,55]. This suggests that a gradual increase in injection rate, where pore pressure is allowed to stabilize between injection stages, may be the safest injection strategy.

**Summary and outlook.** In summary, our model points to the underlying mechanism by which the rate of fluid pressurization, and hence the rate of effective normal stress unloading, may explain several injection-induced seismicity observations[1,11–14,16–18]. An abrupt or large increase in injection rate tends to intensify the early-time destabilizing effect of the rate of change in pore pressure on frictional sliding, whereas a gradual or small increase in injection rate tends to lessen it. Our findings, as a whole, suggest injection strategies to mitigate the seismic risk associated with a wide range of subsurface operations, from wastewater injection to geologic $CO_2$ sequestration. Of course, a complex interplay of different mechanisms such as heterogeneous fault stresses, stress changes from aseismic slip, spatial growth of pore pressure diffusion, and static and dynamic stress transfer often play a role in the occurrence and, in particular, the timing of an earthquake[4]. This emphasizes

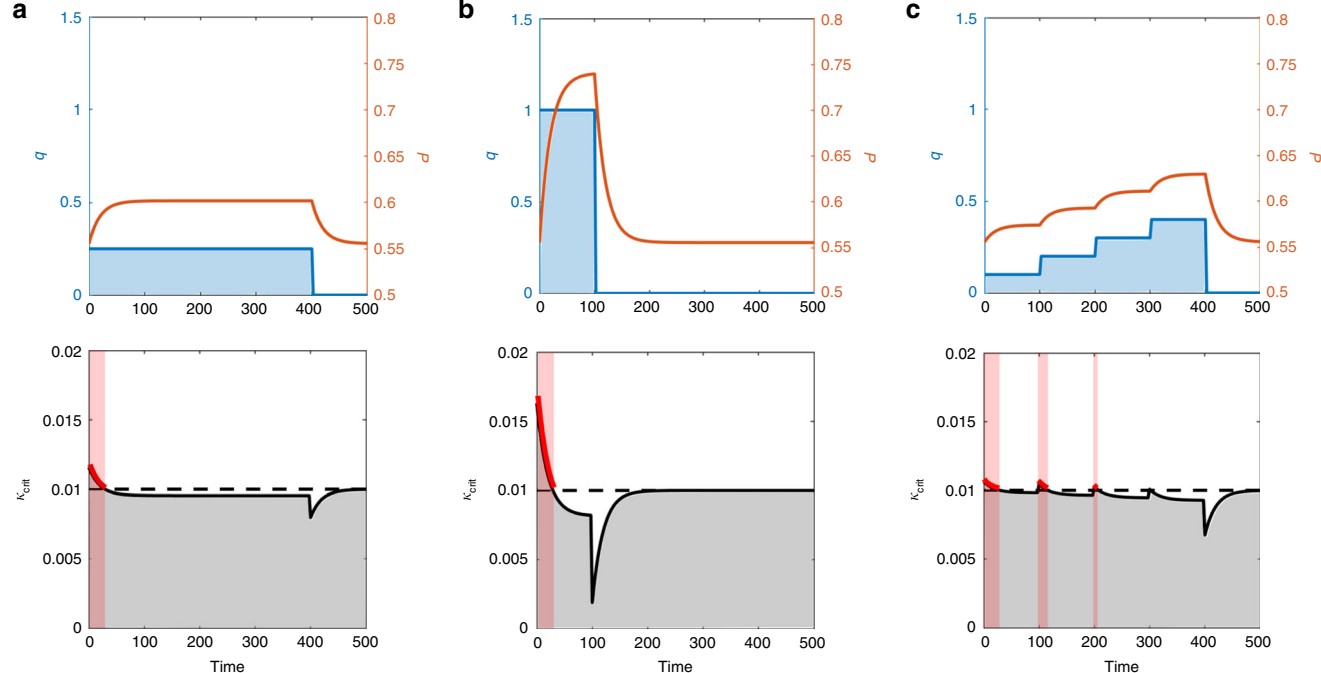

**Fig. 3 Comparison of stability profiles for three different injection scenarios with the same total injected volume. a** reference injection rate and period of injection ($c = 3 \times 10^{-2}$, rq $= 5 \times 10^{-3}$); **b** fourfold increase in injection rate, and corresponding fourfold decrease in the period of injection ($c = 3 \times 10^{-2}$, rq $= 2 \times 10^{-2}$); **c** linear ramp-up in injection rate over the same period of injection as the reference case ($c = 3 \times 10^{-2}$, rq $= 2 \times 10^{-3}$). The top figures show injection rates (blue) and pore pressures (orange). The bottom figures show the critical stiffness in response to the different injection scenarios (solid black) compared to the pre-injection critical stiffness value (dashed black). Higher peak in critical stiffness is indicative of higher likelihood for triggering earthquakes.

the need to continue to develop and test new models for the forecast and control of induced seismicity[10].

## Methods
**Methods described in the Supplementary Information**. All methods and data are described in the Supplementary Information, including: (1) Derivation of the poroelastic spring–slider equations; (2) Governing equations in dimensionless form; (3) Linear stability analysis; (4) Nonlinear simulations; (5) Analytical vs. numerical estimates of critical stiffness; (6) Detailed discussion on the quasi-steady-state approximation; (7) Application to the Denver earthquakes; (8) Phase diagram of injection-induced seismicity; and (9) Earthquake likelihood vs. injection duration.

## Data availability
All relevant data are available upon reasonable request from the authors.

## Code availability
The simulation code is available upon reasonable request from the authors.

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

## Acknowledgements

M.A. was supported by a graduate fellowship from Saudi Aramco. R.J. acknowledges funding from ExxonMobil through its membership in the MIT Energy Initiative.

## Author contributions

R.J. designed research; M.A. performed research; M.A. and R.J. analyzed results; and M.A. and R.J. wrote the paper.

## Competing interests

The authors declare no competing interests.
