## [Peer Review File · Nature Communications]

Understanding rate effects in injection-induced earthquakes

Alghannam and Juanes

Summary Review: This manuscript represents a timely and important contribution to the state of the art in physics-based forecasting of induced earthquakes. While the results are generally intuitive based upon previous models of injection-induced earthquakes (i.e.), this work seeks to better elucidate the physics of why the *way* you inject a given injection volume over time is important i.e. that the earthquake rate is proportional to the stressing rate/pressurization rate for fault embedded in a poroelastic medium with a threshold effect. The manuscript explains, with a physical model, the distinction between injection rate and cumulative volume in the likelihood of induced events. The manuscript is generally well-written and thoughtfully organized. I think it should be published with a minor revision outlined below.

- (1) I think the most important revision that could improve the paper is to make a more intuitive diagram of how K_{critical} exceeding the initial K_{critical} relates to an earthquake probability or rate. This would help readers more clearly connect the abrupt injection rate scenario (Scenario B) with the greatest induced earthquake probability. A casual reader might incorrectly assume that Scenario C was actually more hazardous as the *number* of times that the initial critical stiffness is exceeded is greater than in A or B. I believe what the authors are trying to portray is that the *area* of the curve above the initial K_{critical} line is greatest in Scenario B compared with Scenario A and C. Perhaps an additional line of figures in Figure 3 would plot the earthquake probability vs time for each scenario, but there are probably other ways to show this clearly as well.
- (2) A clearer description with more emphasis on $K < K_{\text{crit}}$ yielding unstable sliding and $K > K_{\text{crit}}$ resulting in stable sliding is needed in the text. Perhaps adding these relationships directly to the text or to one of the figures would help readers as well.
- (3) An additional citation of this paper as another good field example of varying injection rates having an effect on induced earthquakes:

Barbour, A. J., Norbeck, J. H., & Rubinstein, J. L. (2017). The effects of varying injection rates in Osage County, Oklahoma, on the 2016 M w 5.8 Pawnee earthquake. *Seismological Research Letters*, 88(4), 1040-1053.

Reviewers' comments:

Reviewer #2 (Remarks to the Author):

The authors derive and analyze a model system for studying stability of fault sliding in response to shear loading and injection of fluid. They do this by modifying the classic spring-slider idealization of a fault in an elastic solid with rate-and-state friction to account for fluid. This has been done before in the context of landslides (e.g., Iverson, 2005) and the authors are encouraged to make connection to that and subsequent studies that include a description of pore pressure evolution and its coupling to fault strength via the effective stress concept. The fluid injection model is minimal (in keeping with the spring-slider idealization) and has the basic ingredients of injection, storage, and loss. The treatment of storage is standard, and loss is idealized using an approximation to Darcy flow (replacing a pore pressure diffusion equation, as studied in Iverson, 2005, with a pressure relaxation ODE, something that has been done in other studies like Segall et al., 2010, where it is termed membrane diffusion).

The authors perform a linear stability analysis of the system response to small perturbations about steady sliding. However, rather than assuming that the fluid (pore pressure) model is also subject to small perturbations about its own steady state, they assume that the pore pressure is linearly increasing during the slip perturbations (thus keeping a term proportional to pressurization rate). This results in a set of equations for slip dynamics with a forcing term proportional to pressurization rate. The stability analysis of the slip dynamics then reveals an increased tendency for instability that can be traced back to this pressurization rate term. In fact, the system becomes more unstable as pressurization rate is increased, leading the authors to suggest that injection rate during oil and gas and other energy operations might be a contributing factor to controlling the occurrence of unstable slip and induced earthquakes. The possible relevance to injection-induced seismicity, and the link to operational parameters like injection schedule, is what raises this study to the level of possible publication in Nature Communications. However, I believe there are possibly fundamental issues with the analysis procedure, specifically the quasi-steady state approximation, that might invalidate the key result of this study (the destabilizing effect of pressurization rate), so I cannot recommend publication without additional information and justification of this approximation.

Let me first offer an alternative explanation for the instability that the authors have identified. Start with the dimensional state evolution equation (S13). The authors treat the \dot{p} term on the RHS as constant, such that the state variable at steady state depends on \dot{p} . If this steady state is inserted into the friction coefficient in (S12) then what results is an expression for the steady state friction coefficient:

$$\mu_{ss}(v) = \mu^* + (a-b) \ln(v/v^*) + (\alpha \cdot dc/v) \cdot \dot{p} / (\sigma - p).$$

(Apologies for not including $\hat{\{}}$; add those as necessary.) The origin of the instability becomes clear when examining this expression. In addition to the usual logarithmic dependence on slip velocity v , there is another term that is inversely proportional to v . This term is velocity weakening in character and hence potentially destabilizing. One can then linearize this expression to define an effective $a-b$ that is smaller than $a-b$ (i.e., more velocity-weakening). The stability problem with this steady state is then identical to the usual rate-and-state problem without pore pressure evolution, so the standard linearized stability analysis (e.g., Rice et al., 2001) can be applied directly. In fact, if one then defines an effective $a-b$ value using the expressions in section 2.1 of Rice et al., 2001, then the stability results of the present study are immediately obtained.

The question now is whether or not the $\mu_{ss}(v)$ should have the additional velocity-weakening term or not. I suspect not, simply on the intuitive basis that pressurization would not introduce a very strong velocity-weakening behavior, with friction coefficient diverging to positive infinity as slip velocity goes to zero. But to examine this more carefully requires examining the quasi-steady

state assumption that $\dot{p}/(\sigma-p)$ can be regarded as a constant. To address this, I think it is useful to rewrite the state evolution equation with each term being dimensionless (but with dependent variables still being dimensional). In other words, write (S13) as

$$d_c \dot{\theta} / v = -(\theta + b \ln(v/v^*)) + (\alpha d_c / v) \dot{p} / (\sigma-p).$$

I suspect that the quasi-steady state approximation is justified when the time scale of friction and slip dynamics is much smaller than the pore pressure evolution time scale. I believe this means that the dominant balance in the equation above is between the left side and the first term on the right side, with the second term on the right being much smaller. Then it seems that one would freeze not $\dot{p}/(\sigma-p)$, but $(\alpha d_c / v) \dot{p} / (\sigma-p)$, including the v . Then the resulting equation would be solved for steady state θ , which would give the usual steady state friction coefficient without the additional velocity-weakening term (the extra term would simply be a constant added to μ^*). Small perturbations would then be added about this steady state, and the results would agree with the usual stability analysis (without the additional \dot{p} term). This is a loose justification, and I encourage the authors to provide a rigorous derivation and justification for the quasi-steady state approximation in the context of singular perturbation theory. Probably there is a dimensionless parameter, I suspect c or perhaps something involving c and α , that must be much smaller than unity to justify the approximation.

Even if the authors justify the approximation and their mathematical and simulation results are correct, I feel that additional steps must be taken for this study to rise to the level of Nature Communications. Specifically there is presently a large gap between the analysis of this highly idealized model and the real world (or continuum problem of injection adjacent to a fault that might be surrounded by a permeable damage zone). The authors need to close that gap by identifying dimensionless parameter choices that correspond to real-world situations of relevance. In particular, the destabilizing term leads to $b \rightarrow b + \alpha (d_c/v_0) \dot{p}/(\sigma-p)$, so to be of significance the authors must show that $\alpha (d_c/v_0) \dot{p}/(\sigma-p)$ is not vastly smaller than $b = O(0.01)$. But this additional term is exactly the ratio of the friction/slip evolution time scale to the pressure evolution time scale, which by assumption, must be much smaller than unity. It seems like even if this destabilizing effect is real, it is very small compared to the usual destabilizing effect of velocity-weakening friction. If this is the case, then the implications for changes in fault stability as a function of injection rate are overstated.

Despite my concerns above, I do think this is a very important study and nice idealization of the coupled dynamics of pore pressure evolution and fault slip. It is worth publishing after addressing the concerns above. I also have a few additional comments below:

1. Equation (1) defines slip in a nonstandard way. Slip δ is defined via $\dot{\delta} = v$, that is, as the time integral of slip velocity. The variable u is the relative displacement between the load point and the slider. The authors are encouraged to change notation or terminology. Note that the equations are correct as written in the paper, it is just that u should not be called slip.
2. Equation (2) uses a mass times acceleration term for inertia. This is appropriate for systems where the solid material subject to imbalanced forces is of finite extent, as occurs for a thin elastic layer sliding on a rigid substrate, or for a lab experiment. The usual treatment of inertia for a semi-infinite or infinite solid is the radiation damping approximation, where the mass being accelerated expands between the shear wave front (e.g., Rice, 1993). I doubt this will change any of the conclusions of the study, since the linear stability analysis is carried out for the quasi-static system anyway.
3. The main text, around (5), should explain the separation of time scales and use of quasi-steady state approximation (at least conceptually). This was confusing to me when \dot{p} appeared as a "constant" in the stability criterion. Also, it is apparent that perturbations about the true steady state (i.e., steady state with pore pressure being steady state, too) must lead to a more complex characteristic equation and hence stability condition. By the way, have the authors considered that case, too? What is found? From reading Iverson, 2005, accounting for pore pressure diffusion can

have rather complex effects.

4. Below (5) it was stated that p and \dot{p} were dimensionless, but the authors had not previously explained the nondimensionalization; when introduced in (4), p was dimensional.
5. I found it difficult when reading the main text to understand how the concept of normal stiffness would generalize to the real Earth. But my understanding when working through the model derivation is that it is related to the storage properties of the porous system. Perhaps using that language would be more intuitive.
6. Check that all figures in the main text showing critical κ are done under conditions where the quasi-steady state approximation is justified. Mark regions where it is invalid as such.
7. There have been several more recent studies of the Linker-Dieterich effect. One recent one is Kilgore et al., 2012. That study implies a very specific value of α .
8. Appendix section 1.2. I think there might be errors or typos in the derivation of the pore pressure equation. First I was confused by the definition of m_{in} and m_{out} . I would have thought $\dot{m}_{in} = q$, and \dot{m}_{out} would be the Darcy outflow. The difference of these would then be set equal to the storage term $\rho \, dV_f/dt$. If this is right, then (S6) should not have the $-q \, dt$ term. Also the Darcy flux equation (S7) should have only m_{out} on the left side. Additionally (S7) is also missing a factor of the fault cross-sectional area, unless permeability is really permeability times cross-sectional area. The cross-sectional area is also missing in (S8). Finally, I think density is missing in (S10).
9. The analysis might be simplified if the reference velocity v^* was chosen to be v_0 , without loss of generality.
10. State the total number of dimensionless parameters in the system (S15)-(S18) and their conceptual meaning. Have the authors studied the role of each parameter? Maybe this is not necessary. But it is very important to state and justify the realistic range of values of the dimensionless parameters, especially to make connection to the real world and real injection schedules. Here it will be important to explain the role of the diffusion length L as well as the normal stiffness k_n (the latter might be connected to storage properties of a continuum).
11. Figure S1. Adaptive step size, mentioned in caption, is not shown. Also, slip shown here is probably the usual one (time integral of v) since it increases linearly with time for stable sliding (whereas u would be constant).
12. Figure S2. Is the change in κ_{cr} coming from the change in effective stress (i.e., the usual dependence of critical stiffness on effective stress)? Or it is really being caused by the additional \dot{p} term?
13. The authors are encouraged to connect their results with a continuum linear stability analysis (and nonlinear simulations) with rate-and-state friction in a poroelastic medium by Heimisson et al., 2019. There have also been a few papers showing nonlinear simulations with rate-and-state friction and poroelasticity, including injection: Torberntsson et al., 2018; Pampillon et al., 2018; maybe also by Chunfang Meng at MIT?

Heimisson, E. R., Dunham, E. M., & Almquist, M. (2019). Poroelastic effects destabilize mildly rate-strengthening friction to generate stable slow slip pulses. *Journal of the Mechanics and Physics of Solids*.

Iverson, R. M. (2005). Regulation of landslide motion by dilatancy and pore pressure feedback. *Journal of Geophysical Research: Earth Surface*, 110(F2).

Kilgore, B., Lozos, J., Beeler, N., & Oglesby, D. (2012). Laboratory observations of fault strength in response to changes in normal stress. *Journal of Applied Mechanics*, 79(3), 031007.

Pampillón, P., Santillán, D., Mosquera, J. C., & Cueto-Felgueroso, L. (2018). Dynamic and quasi-dynamic modeling of injection-induced earthquakes in poroelastic media. *Journal of Geophysical Research: Solid Earth*, 123(7), 5730-5759.

Rice, J. R. (1993). Spatio-temporal complexity of slip on a fault. *Journal of Geophysical Research: Solid Earth*, 98(B6), 9885-9907.

Segall, P., Rubin, A. M., Bradley, A. M., & Rice, J. R. (2010). Dilatant strengthening as a mechanism for slow slip events. *Journal of Geophysical Research: Solid Earth*, 115(B12).

Torberntsson, K., Stiernström, V., Mattsson, K., & Dunham, E. M. (2018). A finite difference method for earthquake sequences in poroelastic solids. *Computational Geosciences*, 22(5), 1351-1370.

Reviewer #3 (Remarks to the Author):

This manuscript proposes a rather schematic analytical model of an injection-induced earthquake on a tectonically loaded fault.

The basis of the consideration is a system of equations describing the rate and state phenomenology of the friction coefficient. Another part of the equation system attempts to account for the poroelastic coupling and the impact of the pore pressure diffusion due to injection.

The authors of the manuscript analyse then the stability / instability trade off of their resulting very non-linear (and thus extremely sensitive to many parameter perturbations) equation system in terms of the critical stiffness of the system.

When the system gets unstable, they claim an earthquake.

I have not found any clear poroelastic coupling formulation. Such one should take into account mutual impacts of stresses and pressures. This is not the case here. Also the assumption of non-compressibility of the fluid seems to be strange in this particular case. Seismic waves have usually velocities up to 5 times higher than the acoustic waves in the water. Thus, the fluid is significantly more compressive than the rock.

Also equations shown in 1-5 are quite unclear. Are they all dimensionless? However it is not really defined. Characteristic normalizing quantities are not defined. The pore pressure and the pressure inside slider are indistinguishable etc..

Beside of these details making the paper badly readable it is completely unclear what would justify its high scientific significance, innovative character and relation to the reality.

For example, the fact that the injection rate (thus, pressure rate) are governing the seismicity rate and thus, the probability of strong earthquakes is very well known.

Corresponding theoretical considerations for linear and non-linear poroelastic systems clearly supported by

real observations can be found e.g., in Shapiro, Dinske, 2009, *JGR*, v. 114, B9.

Also the claims of the manuscript of their result similarity with some case studies (such a similarity is not really documented in the manuscript) are not valid. For example, the Basel 2006 event of Mw3.2 (Majer et al, 2007, *Geothermics*, v.36, 185-222) occurred exactly by the type of the injection shown in the sketch 3 C of the manuscript (recommended by the authors as a stabilizing strategy!). Moreover, the injection was stopped because significant events of Mw close to 3 started to happen and not, because the injection was stopped, the events started to happen (as Fig 3 implies). Also in Pohang, the Mw5.5 occurred already 2 months after the injection was

stopped. Moreover, it happened after a cycle with a smallest injection rate.

UNDERSTANDING RATE EFFECTS IN INJECTION-INDUCED EARTHQUAKES.

Amendments in response to the comments from Referee 1.

We thank the referee for a thoughtful and positive evaluation of the manuscript. He states that our study “represents a timely and important contribution to the state of the art in physics-based forecasting of induced earthquakes”, and that the manuscript is “well-written and thoughtfully organized”. He concludes that our study “should be published with a minor revision”. In what follows, we respond to each of the referee’s comments in turn. The referee’s concerns are included verbatim in *italics* and our responses follow each point.

Comment 1. I think the most important revision that could improve the paper is to make a more intuitive diagram of how $K_{critical}$ exceeding the initial $K_{critical}$ relates to an earthquake probability or rate. This would help readers more clearly connect the abrupt injection rate scenario (Scenario B) with the greatest induced earthquake probability. A casual reader might incorrectly assume that Scenario C was actually more hazardous as the number of times that the initial critical stiffness is exceeded is greater than in A or B. I believe what the authors are trying to portray is that the area of the curve above the initial $K_{critical}$ line is greatest in Scenario B compared with Scenario A and C. Perhaps an additional line of figures in Figure 3 would plot the earthquake probability vs time for each scenario, but there are probably other ways to show this clearly as well.

Response. We thank the referee for raising an interesting and practical point. In Fig. 3 of the original manuscript, we are trying to portray that the peak in critical stiffness is greatest in scenario B, which correlates to the greatest earthquake likelihood. To clarify this point, we have modified Fig. 3 (Fig. 1 below) plots and caption to emphasize the peak critical stiffness and added a plot showing the likelihood of earthquakes as a function of the duration of the injection period for a fixed total injected volume (Fig. 2 below). As it is now clear, the earthquake likelihood, inferred from the peak critical stiffness, is directly correlated with the length of the injection period. A shorter injection duration or equivalently a higher injection rate results in a higher likelihood of earthquake triggering.

To address this point, we have updated Fig. 3 (Fig. 1 below) in the main text and we have added section 9 in the supplementary materials that discusses the earthquake likelihood as a function of injection duration.

FIGURE 1. Comparison of stability profiles for three different injection scenarios with the same total injected volume: (A) reference injection rate and period of injection ($c = 3 \times 10^{-2}$, $rq = 5 \times 10^{-3}$); (B) four-fold increase in injection rate, and corresponding four-fold decrease in the period of injection ($c = 3 \times 10^{-2}$, $rq = 2 \times 10^{-2}$); (C) linear ramp-up in injection rate over the same period of injection as the reference case ($c = 3 \times 10^{-2}$, $rq = 2 \times 10^{-3}$). The top figures show injection rates (blue) and pore pressures (orange). The bottom figures show the critical stiffness in response to the different injection scenarios (solid black) compared to the pre-injection critical stiffness value (dashed black). Higher peak in critical stiffness is indicative of higher likelihood for triggering earthquakes.

FIGURE 2. Earthquake likelihood vs. injection duration. The earthquake likelihood is computed as a normalized initial critical stiffness for a given injection duration ($\kappa_{\text{crit}}|_T / \kappa_{\text{crit}}|_{T_{\text{min}}}$). The injection duration is computed as the ratio of a fixed total injection volume to the injection rate ($T = V/q$).

Comment 2. *A clearer description with more emphasis on $K < K_{crit}$ yielding unstable sliding and $K > K_{crit}$ resulting in stable sliding is needed in the text. Perhaps adding these relationships directly to the text or to one of the figures would help readers as well.*

Response. To clarify this point and another point raised by another reviewer about the quasi-steady-state approximation, we have modified the paragraph around κ_{crit} in the main text to (line 94):

The stability of steady frictional sliding to small perturbations in velocity, which determines whether motion is by slow steady-sliding or violent stick-slip, depends on the evolution of the frictional resistance. Stick-slip occurs whenever a change of frictional resistance with sliding occurs at a rate greater than the loading system is capable of following [1]. At a constant pore pressure, linear stability analysis of the system about steady-state leads to the stability condition by Ruina (1983) [2]. Pore pressure, however, is not constant in time and its evolution depends on the injection rate and on the poroelastic and hydraulic parameters of the rupture. To quantify this, we carry out a linear stability analysis of the system about a quasi steady-state where sliding is steady but pore pressure is evolving as a result of fluid injection. We find that motion is by stick-slip when the dimensionless shear stiffness of the loading system is lower than a critical value ($\kappa < \kappa_{crit}$) given by

$$\kappa_{crit} = (b - a)(\sigma - p) + \frac{\alpha}{v_0}\dot{p}, \quad (1)$$

and is by steady-sliding otherwise ($\kappa > \kappa_{crit}$). Variables p and \dot{p} are dimensionless pore pressure magnitude and pore pressure rate, respectively, at any point in time. Accordingly, frictional instability for the spring–poroslider system with an evolving pore pressure depends not only on the magnitude of pore pressure, but also on the rate of change of pore pressure (see supplementary materials for the analysis of QSSA, derivation of Eq. (1), and validation against nonlinear simulations).

We have also added these relationships to the caption of Figure 2A in manuscript:

Dimensionless critical stiffness and critical injection rate. (A) Critical stiffness—below which stick-slip instability is triggered ($\kappa < \kappa_{crit}$) and above which stick-slip instability is inhibited ($\kappa > \kappa_{crit}$)—in response to fluid injection at constant rate in velocity-weakening material ($c = 3 \times 10^{-2}$, $rq = 5 \times 10^{-3}$). The combined effect is illustrated by the solid curve, which shows a transition in the effect of fluid injection from de-stabilizing at early times to stabilizing at late times. The effect of the magnitude of pore pressure is illustrated by the dotted curve. The effect of the rate of change in pore pressure is illustrated by the dashed curve. (B) Critical injection rate—above

which stick-slip instability is triggered—as a function of the normalized diffusivity of the system. The diffusivity is varied by varying permeability (beige curve in B).

Comment 3. *An additional citation of this paper as another good field example of varying injection rates having an effect on induced earthquakes:*

Barbour, A. J., Norbeck, J. H., & Rubinstein, J. L. (2017). The effects of varying injection rates in Osage County, Oklahoma, on the 2016 Mw 5.8 Pawnee earthquake. Seismological Research Letters, 88(4), 1040-1053.

Response. We thank the referee for pointing us to this recent paper on rate effects on induced earthquakes. We have added a reference to it in the introduction [3] (line 40).

In summary, we thank the referee for the thoughtful comments, which have helped improve our manuscript. We hope that our response has properly addressed the points raised in his review.

REFERENCES

- [1] JD Byerlee. The mechanics of stick-slip. *Tectonophysics*, 9(5):475–486, 1970.
- [2] Andy Ruina. Slip instability and state variable friction laws. *Journal of Geophysical Research: Solid Earth*, 88(B12):10359–10370, 1983.
- [3] Andrew J Barbour, Jack H Norbeck, and Justin L Rubinstein. The effects of varying injection rates in Osage County, Oklahoma, on the 2016 M_w 5.8 Pawnee earthquake. *Seismological Research Letters*, 88(4):1040–1053, 2017.

UNDERSTANDING RATE EFFECTS IN INJECTION-INDUCED EARTHQUAKES.

Amendments in response to the comments from Referee 2.

We thank the referee for a thoughtful and positive evaluation of the manuscript. He states that our study is “very important” and “nice idealization of the coupled dynamics of pore pressure evolution and fault slip”. He offers an accurate summary and re-interpretation of the main analytical results of our paper. He concludes that our study “is worth publishing after addressing the concerns”. In what follows, we respond to each of the referee’s concerns in turn. The referee’s comments are included verbatim in *italics* and our responses follow each point.

Major comment 1. The question now is whether or not the $\mu_{ss}(v)$ should have the additional velocity-weakening term or not. I suspect not, simply on the intuitive basis that pressurization would not introduce a very strong velocity-weakening behavior, with friction coefficient diverging to positive infinity as slip velocity goes to zero.

Response. The referee raises an interesting point. Olsson (1988) [1] performed laboratory tests in which the normal stress was increased at constant rate while the load point speed was held constant. He found that shear stress is a function of the normal stress rate. When the normal stress rate was increased by 10 during steady sliding, the rate of increase of shear stress with normal stress (coefficient of friction) decreased by a factor of two. This seems to be a significant effect. We have included this point in section 1.1 of the supplementary materials (line 34).

Major comment 2. I encourage the authors to provide a rigorous derivation and justification for the quasi-steady state approximation in the context of singular perturbation theory. Probably there is a dimensionless parameter, I suspect c or perhaps something involving c and α , that must be much smaller than unity to justify the approximation.

Response. We have taken this comment very seriously and made every effort to address it fully. The quasi-steady-state approximation, in general, is an approach to simplify dynamic systems of ordinary differential equations with an initial fast transient, after which some of the dependent variables can be assumed to be in steady-state with regard to the other slowly evolving dependent variables [2]. In particular, the QSSA is a good approach to use in our analysis because it allows us to study the stability of steady frictional sliding to small perturbations in velocity while pore pressure is evolving. The sliding velocity and the velocity-dependent part of the state variable are in steady state with respect to the pore pressure. Here, we analyze the QSSA in the context of singular perturbation theory

following the analysis by Segel and Slemrod (1989) [2], identify the small parameter(s) necessary for the validity of the QSSA, and quantify the error associated with it.

1. REDUCED DIMENSIONAL EQUATIONS

As shown in section 1, the dynamics of our poroelastic spring-slider model is governed by a system of four coupled nonlinear ODEs. Under quasi-static loading, velocity is the fastest evolving variable of the system and it responds instantaneously (negligible inertia) to small perturbations. Thus we can focus our analysis on a reduced system of ODEs at steady-state velocity $V = V_0$,

$$\dot{\Theta} = -\frac{V_0}{d_c}(\Theta + \hat{b} \ln \frac{V_0}{V_*}) + \hat{\alpha} \frac{\dot{P}}{\Sigma - P}, \quad (1)$$

$$\dot{P} = \frac{k_n^{\text{eff}} k}{\eta L} (P_0 - P) + k_n^{\text{eff}} Q, \quad (2)$$

with initial conditions

$$\Theta(0) = 0, \quad (3)$$

$$P(0) = P_0. \quad (4)$$

2. TIMESCALES

As a first step in the analysis, we estimate the fast timescale t_Θ of the pre-steady-state period and the slow timescale t_P for the evolution of pore pressure. To estimate t_Θ we make the approximation $P \approx P_0$ in Eq. (1). The solution for the state variable becomes

$$\Theta(t) = \bar{\Theta}(e^{-\frac{V_0}{d_c}t} - 1), \quad (5)$$

where $\bar{\Theta} = \hat{b} \ln V_0/V_* - \hat{\alpha}(d_c/V_0)(\dot{P}_0/(\Sigma - P_0))$. Subsequently, we take

$$t_\Theta = \frac{d_c}{V_0}. \quad (6)$$

Since the pore pressure evolution is independent of the evolution of the state variable Θ , we estimate t_P by solving Eq. (2) with the initial condition $P(0) = P_0$ to obtain

$$P(t) = P_0 + \frac{\eta L}{k} Q(1 - e^{-\frac{k_n^{\text{eff}} k}{\eta L}t}). \quad (7)$$

Similarly, we take

$$t_P = \frac{\eta L}{k_n^{\text{eff}} k}. \quad (8)$$

3. SCALED DIMENSIONLESS EQUATIONS

During the pre-steady-state, it is reasonable to scale time by t_Θ , where the dimensionless time τ is given by

$$\tau = \frac{t}{t_\Theta}. \quad (9)$$

Thus, the scaled dimensionless governing equations become

$$\frac{\partial \theta}{\partial \tau} = -v_0(\theta + b \ln v_0) + \frac{\alpha}{\sigma - p} \frac{\partial p}{\partial \tau}, \quad (10)$$

$$\frac{\partial p}{\partial \tau} = \frac{t_\Theta}{t_P}(p_0 - p) + t_\Theta q, \quad (11)$$

with initial conditions

$$\theta(0) = 0, \quad (12)$$

$$p(0) = p_0, \quad (13)$$

where we have defined $b = \hat{b}/\mu_0$, $\alpha = \hat{\alpha}/\mu_0$, and $q = Q/p_0$. After the pre-steady state, the QSSA is assumed to be valid and t_P becomes a reasonable timescale. We introduce a new dimensionless scaled time T by

$$T = \frac{t}{t_P}, \quad (14)$$

with which the scaled dimensionless governing equations become

$$\frac{t_\Theta}{t_P} \frac{\partial \theta}{\partial T} = -v_0(\theta + b \ln v_0) + \frac{t_\Theta}{t_P} \frac{\alpha}{\sigma - p} \frac{\partial p}{\partial T}, \quad (15)$$

$$\frac{\partial p}{\partial T} = (p_0 - p) + t_P q. \quad (16)$$

4. SINGULAR PERTURBATION

Approximate solutions can now be obtained by methods of singular perturbation theory [3], for $0 < t_\Theta/t_P \ll 1$. A solution of Eqs. (10)-(11) is obtained of the form

$$\theta(\tau) = \theta^{(0)}(\tau) + \frac{t_\Theta}{t_P} \theta^{(1)}(\tau) + \dots, \quad (17)$$

$$p(\tau) = p^{(0)}(\tau) + \frac{t_\Theta}{t_P} p^{(1)}(\tau) + \dots, \quad (18)$$

where

$$\theta^{(0)}(\tau) = \bar{\theta}(e^{-\tau} - 1), \quad (19)$$

$$p^{(0)}(\tau) = p_0. \quad (20)$$

Similarly, the solution of Eqs. (15)-(16) obtained of the form

$$\theta(T) = \theta_0(T) + \frac{t_\Theta}{t_P} \theta_1(T) + \dots, \quad (21)$$

$$p(T) = p_0(T) + \frac{t_\Theta}{t_P} p_1(T) + \dots, \quad (22)$$

where

$$\theta_0 = -b \ln v_0 + \frac{\alpha}{v_0} \frac{t_\Theta q}{\sigma - p_0}, \quad (23)$$

$$\frac{\partial p_0}{\partial T} = t_P q. \quad (24)$$

Note that Eqs. (23)-(24) are associated with initial conditions Eqs. (3)-(4), and represent the initial state about which we linearized the spring-porosluder system (section 3). The results of this analysis remain valid for general initial conditions $\Theta(0) = \Theta_i$ and $P(0) = P_i$ [2], where Θ_i ranges from 0 to Θ_{ss} and P_i ranges from P_0 to P_{ss} . Linearizing the spring-porosluder system about the true steady-state, where

$$\theta_0 = -b \ln v_0, \quad (25)$$

$$\frac{\partial p_0}{\partial T} = 0, \quad (26)$$

thus yields the Ruina (1983) stability condition [4].

5. QSSA VALIDITY CONDITIONS

A necessary aspect of the QSSA is that the duration of the pre-steady-state period is much shorter than the characteristic time for the pore pressure evolution. An essential condition for the QSSA to be valid after the pre-steady state is therefore $t_\Theta \ll t_P$,

$$0 < c = \left(\frac{d_c}{V_0} \right) / \left(\frac{\eta L}{k_n^{\text{eff}} k} \right) \ll 1. \quad (27)$$

Note that the initial condition $P(0) = P_0$ is reasonable for the QSSA only if there is a negligible relative change $|\Delta P/P_0|$ in pore pressure during the pre-steady state. We estimate $|\Delta P/P_0|$ by

$$\left| \frac{\Delta P}{P_0} \right| \approx \frac{1}{P_0} \left| \frac{\partial P}{\partial t} \right|_{\text{max}} t_\Theta. \quad (28)$$

An additional condition for the validity of the QSSA is therefore

$$0 < r q = \frac{d_c k_n^{\text{eff}}}{V_0} \frac{Q}{P_0} \ll 1. \quad (29)$$

Recall that parameter c is the normalized diffusivity or ratio of the pore pressure to the state evolution timescales and parameter $r q$ is the normalized injection rate, where $r = t_\Theta k_n^{\text{eff}}$ and $q = Q/P_0$.

6. ERROR ESTIMATES

This analysis shows that using the QSSA to study the stability of steady frictional sliding to small perturbations in velocity with an evolving pore pressure is justified when conditions Eqs. (27) and (29) are met. In other words, if in a timescale t_Θ sliding reaches steady state with a constant pore pressure, then assuming that sliding is in a quasi-steady state with a changing pore pressure is valid when the pore pressure change occurs on a time scale t_P that is long compared to t_Θ and $\Delta P|_{t_\Theta}$ is small compared to P_0 .

Therefore, we expect that the accuracy of our instability criterion depends on dimensionless parameters c and rq . Here we evaluate the error in the analytical estimate of the critical stiffness required to trigger the first slip event (Figure 1). We indeed find that the error decreases as c or rq decrease. It becomes small ($< 15\%$) when the normalized diffusivity and normalized injection rate reach small values ($c \leq 5 \times 10^{-2}$, $rq \leq 5 \times 10^{-3}$). It is also interesting to note that the QSSA validity may be extended to instances where c is of order one provided that rq is significantly smaller than one (green curve).

FIGURE 1. Error in the instability criterion as a function of c and rq . The error is taken as the ratio of the maximum difference between the analytical and numerical estimates of the critical stiffness required to trigger the first slip event to the numerical estimate. The dimensionless parameter c is varied by varying permeability k , and rq is varied by varying injection rate Q .

To address this comment, we have added an entire new section of the supplementary materials (section 6) and added a reference to it in the main text (line 108).

Major comment 3. *There is presently a large gap between the analysis of this highly idealized model and the real world (or continuum problem of injection adjacent to a fault that might be surrounded by a permeable damage zone). The authors need to close that gap by identifying dimensionless parameter choices that correspond to real-world situations of relevance. In particular, the destabilizing term leads to $b- > b + \alpha(d_c/v_0)\dot{p}/(\sigma - p)$, so to be of significance the authors must show that $\alpha(d_c/v_0)\dot{p}/(\sigma - p)$ is not vastly smaller than $b = O(0.01)$. But this additional term is exactly the ratio of the friction/slip evolution time scale to the pressure evolution time scale, which by assumption, must be much smaller than unity. It seems like even if this destabilizing effect is real, it is very small compared to the usual destabilizing effect of velocity-weakening friction. If this is the case, then the implications for changes in fault stability as a function of injection rate are overstated.*

Response. We thank the referee for raising this important point. To bridge the gap between the analysis of the idealized spring-porosluder model and the real world, we express our instability criterion in dimensional form, and identify values of dimensionless parameters c and rq that correspond to real-world settings.

In dimensional form, the critical stiffness from the linear stability analysis is

$$k_{s,\text{crit}} = \frac{(\hat{b} - \hat{a})}{d_c}(\Sigma - P) + \frac{\hat{\alpha}}{V_0}\dot{P}, \quad (30)$$

or, equivalently,

$$k_{s,\text{crit}} = \left[(\hat{b} - \hat{a}) + \hat{\alpha} \frac{d_c}{V_0} \frac{\dot{P}}{(\Sigma - P)} \right] \frac{(\Sigma - P)}{d_c}, \quad (31)$$

where the term $\hat{b} - \hat{a}$ represents the original velocity weakening effect and the dimensionless term $\hat{\alpha}(d_c/V_0)\dot{P}/(\Sigma - P)$ represents an additional weakening effect from fluid pressurization. Note that this pressurization term is maximum at early times and is approximately equal to rq .

The 1960s Denver earthquakes is a good example of a real-world setting, where it is well-documented that injection of wastewater into the fractured Precambrian granite gneiss underneath the Rocky Mountain Arsenal triggered the earthquakes and where injection rate is directly related to the frequency of earthquakes [5, 6]. The reservoir spans a depth interval from 3.7 to 7 km below the surface. Experimental data on granite at this depth shows velocity weakening behavior ($\hat{b} - \hat{a}$ in the range 0.002 to 0.005, $\mu_0 = 0.7$ to 0.75) [7].

To identify values of dimensionless parameter c that correspond to this setting, we estimate the state evolution timescale t_Θ and the pore pressure evolution timescale t_p :

$$c = \frac{t_\Theta}{t_p}. \quad (32)$$

The state evolution timescale t_Θ is

$$t_\Theta = \frac{d_c}{V_0}. \quad (33)$$

We find that t_Θ ranges from 10 days to 4 months based on field data of the characteristic slip distance and loading rate ($d_c = 10^{-3}$ to 10^{-2} m, $V_0 = 10^{-9}$ m/s) [8, 9].

The pore pressure evolution time scale t_P can be transferred from the poroslider model to field settings,

$$t_P = \underbrace{\frac{\eta L}{k_n^{\text{eff}} k}}_{\text{poroslider}} = \frac{L}{k/\eta} \left[\frac{1}{k_n} + H_0 c_f \right] = \frac{L^2}{k/\eta} \left[\frac{1}{K_v} + \phi \frac{1}{K_f} \right] = \frac{L^2}{k/\eta} \frac{S_s}{\rho g} = \underbrace{\frac{L^2 S}{T}}_{\text{field}}. \quad (34)$$

We find that t_P is approximately 2 years based on a reservoir analysis of the Denver earthquakes, with transmissivity $T = 10^{-5}$ m²/s, storativity $S = 10^{-5}$, and characteristic length scale $L = 8 \times 10^3$ m [10].

In a similar manner, we identify values of dimensionless parameter rq . We translate this quantity from the poroslider model to field settings:

$$rq = \underbrace{\frac{d_c}{V_0} k_n^{\text{eff}} \frac{Q}{P_0}}_{\text{poroslider}} = \frac{d_c}{V_0} k_n^{\text{eff}} \frac{Q_w}{P_0 W B} = \frac{d_c}{V_0} \frac{\left[\frac{1}{K_v} + \phi \frac{1}{K_f} \right]^{-1}}{L} \frac{Q_w}{P_0 W B} = \underbrace{\frac{d_c \rho g}{V_0 L S} \frac{Q_w}{P_0 W}}_{\text{field}}, \quad (35)$$

and evaluate values based on the reservoir analysis and injection data, with reservoir pressure $P_0 = 30$ MPa, reservoir width $W = 3 \times 10^3$ m, and field injection rate $Q_w = 2$ to 9 million gal/mo [5, 10].

Therefore, reasonable estimates of c and rq for this setting would be in the order of 10^{-2} to 10^{-1} and 10^{-3} to 10^{-1} , respectively. Note that both estimates are much smaller than one, and thus meet the QSSA validity conditions.

Having determined the validity of the QSSA analysis to this setting, we now assess whether pressurization rate effects were likely significant during fluid injection leading to the Denver earthquakes. In dimensionless form, the critical stiffness κ_{crit} is given by Eq. (S29). Prior to fluid injection, the pore pressure is constant and the critical stiffness,

$$\kappa_{\text{crit}} = (b - a)(\sigma - p_0), \quad (36)$$

is estimated to be 0.005 ($b - a = 0.003$ to 0.007 , $\sigma - p_0 = 1$). Shortly following the start of fluid injection, the pore pressure increases rapidly and the dimensionless critical stiffness takes the form:

$$\kappa_{\text{crit}} \Big|_{t=0} = (b - a)(\sigma - p_0) + \frac{\alpha}{v_0} \dot{p} \Big|_{t=0} = (b - a)(\sigma - p_0) + \frac{\alpha}{v_0} rq. \quad (37)$$

This results in an increase in critical stiffness at early times in the range of 30% to 3000% ($\alpha = 1$, $v_0 = 1$), thus indicating that the additional weakening effect from fluid pressurization is likely significant in this setting.

To address this point, we have added a new section in the supplementary materials (section 7) and have added a summary in the main text (line 143).

Comment 1. Equation (1) defines slip in a nonstandard way. Slip delta is defined via $\dot{\delta} = v$, that is, as the time integral of slip velocity. The variable u is the relative displacement between the load point and the slider. The authors are encouraged to change notation or terminology. Note that the equations are correct as written in the paper, it is just that

u should not be called slip.

Response. We thank the referee for making this observation. We have changed the terminology for u in the main text from “slip” to “relative displacement between the load point and the slider” (line 72). One reason why we formulated the problem in terms of relative displacement instead of slip is to avoid having t as an explicit variable in Equation (2).

Comment 2. Equation (2) uses a mass times acceleration term for inertia. This is appropriate for systems where the solid material subject to imbalanced forces is of finite extent, as occurs for a thin elastic layer sliding on a rigid substrate, or for a lab experiment. The usual treatment of inertia for a semi-infinite or infinite solid is the radiation damping approximation, where the mass being accelerated expands between the shear wave front (e.g., Rice, 1993). I doubt this will change any of the conclusions of the study, since the linear stability analysis is carried out for the quasi-static system anyway.

Response. The reviewer is correct in that, generally, the mass of rock is not all mobilized instantaneously and, therefore, a radiation-damping term would be appropriate for a semi-infinite medium. Such a term would affect slightly the nonlinear numerical simulations, but—as the reviewer also points out—not the linear stability analysis. Thus, we have decided to keep the description with constant rock mass in the present paper.

Comment 3. The main text, around (5), should explain the separation of time scales and use of quasi-steady state approximation (at least conceptually). This was confusing to me when \dot{p} appeared as a “constant” in the stability criterion. Also, it is apparent that perturbations about the true steady state (i.e., steady state with pore pressure being steady state, too) must lead to a more complex characteristic equation and hence stability condition. By the way, have the authors considered that case, too? What is found? From reading Iverson, 2005, accounting for pore pressure diffusion can have rather complex effects.

Response. We thank the referee for pointing out the confusion, and to the paper by Iverson (2005). We have modified the paragraph around Eq. (5) to (line 94):

The stability of steady frictional sliding to small perturbations in velocity, which determines whether motion is by slow steady-sliding or violent stick-slip, depends on the evolution of the frictional resistance. Stick-slip occurs whenever a change of frictional resistance with sliding occurs at a rate greater than the loading system is capable of following [11]. At a constant pore pressure, linear stability analysis of the system about steady-state leads to the stability condition by Ruina (1983) [4]. Pore pressure, however, is not constant in time and its evolution depends on the injection rate and on the poroelastic and hydraulic parameters of the rupture. To quantify this, we carry out a linear stability analysis of the system about a quasi steady-state where sliding is steady but pore pressure is evolving as a result of fluid

injection. We find that motion is by stick-slip when the dimensionless shear stiffness of the loading system is lower than a critical value ($\kappa < \kappa_{\text{crit}}$) given by

$$\kappa_{\text{crit}} = (b - a)(\sigma - p) + \frac{\alpha}{v_0} \dot{p}, \quad (38)$$

and is by steady-sliding otherwise ($\kappa > \kappa_{\text{crit}}$). Variables p and \dot{p} are dimensionless pore pressure magnitude and pore pressure rate, respectively, at any point in time. Accordingly, frictional instability for the spring–poroslider system with an evolving pore pressure depends not only on the magnitude of pore pressure, but also on the rate of change of pore pressure (see supplementary materials for the analysis of QSSA, derivation of Eq. (38), and validation against nonlinear simulations).

Linearizing the system about the true steady-state and assuming that pore pressure is subject to arbitrary small perturbations about its own steady-state leads to the following system of equations

$$\begin{bmatrix} \Delta \dot{v} \\ \Delta \dot{\theta} \\ \Delta \dot{p} \end{bmatrix} = \begin{bmatrix} \frac{b}{a} v_{\text{ss}} - \frac{\kappa v_{\text{ss}}}{a(\sigma - p_{\text{ss}})} & -\frac{v_{\text{ss}}^2}{a} & \frac{c v_{\text{ss}} \mu_{\text{ss}}}{a(\sigma - p_{\text{ss}})} \\ -b & -v_{\text{ss}} & -\frac{c\alpha}{(\sigma - p_{\text{ss}})} \\ 0 & 0 & -c \end{bmatrix} \begin{bmatrix} \Delta v \\ \Delta \theta \\ \Delta p \end{bmatrix}, \quad (39)$$

and characteristic equation

$$(c + \lambda)(a(\sigma - p_{\text{ss}})\lambda^2 + (\kappa v_{\text{ss}} - (b - a)(\sigma - p_{\text{ss}})v_{\text{ss}})\lambda + \kappa v_{\text{ss}}^2) = 0. \quad (40)$$

Note that $\lambda_1 = -c$, and so this leads to the Ruina (1983) instability condition. Linearizing the system about the true steady-state and assuming that pore pressure is subject to imposed injection-induced perturbations about its own steady state (similar to the approach followed by Iverson (2005) [12]) leads to the following system of equations

$$\begin{bmatrix} \Delta \dot{v} \\ \Delta \dot{\theta} \\ \Delta \dot{p} \end{bmatrix} = \begin{bmatrix} \frac{b}{a} v_{\text{ss}} - \frac{\kappa v_{\text{ss}}}{a(\sigma - p_{\text{ss}})} & -\frac{v_{\text{ss}}^2}{a} & \frac{c v_{\text{ss}} \mu_{\text{ss}}}{a(\sigma - p_{\text{ss}})} \\ -b & -v_{\text{ss}} & -\frac{c\alpha}{(\sigma - p_{\text{ss}})} \\ 0 & 0 & -c \end{bmatrix} \begin{bmatrix} \Delta v \\ \Delta \theta \\ \Delta p \end{bmatrix} + \begin{bmatrix} -\frac{v_{\text{ss}} \mu_{\text{ss}}}{a(\sigma - p_{\text{ss}})} \\ \frac{\alpha}{(\sigma - p_{\text{ss}})} \end{bmatrix} r q. \quad (41)$$

For a constant and positive injection rate q , this also leads to the Ruina (1983) instability condition. A reason behind the complex pore pressure diffusion effect in Iverson (2005) model is that steady-state pore pressure is strongly coupled to steady-state slip velocity through shear-induced dilatancy and contraction, whereas these quantities are independent in our model.

Comment 4. Below (5) it was stated that p and \dot{p} were dimensionless, but the authors had not previously explained the nondimensionalization; when introduced in (4), p was dimensional.

Response. To clarify the confusion, we now use uppercase P for dimensional pore pressure and lowercase p for dimensionless pore pressure in our governing equations in both main manuscript (line 71) and supplementary materials (line 80).

$$\dot{U} = V_0 - V, \quad (42)$$

$$\dot{V} = \frac{1}{(T/2\pi)^2} \left[U - \frac{1}{k_s} (\mu_* + \hat{a} \ln \frac{V}{V_*} + \Theta)(\Sigma - P) \right], \quad (43)$$

$$\dot{\Theta} = -\frac{V}{d_c} (\Theta + \hat{b} \ln \frac{V}{V_*}) + \hat{\alpha} \frac{\dot{P}}{(\Sigma - P)}, \quad (44)$$

$$\dot{P} = \frac{k_n^{\text{eff}} k}{\eta L} (P_0 - P) + k_n^{\text{eff}} Q. \quad (45)$$

We also refer to the supplementary materials for the derivation of the dimensionless form of the equations (section 2), and we have included the dimensionless form of the equations also in the manuscript (line 84).

Comment 5. I found it difficult when reading the main text to understand how the concept of normal stiffness would generalize to the real Earth. But my understanding when working through the model derivation is that it is related to the storage properties of the porous system. Perhaps using that language would be more intuitive.

Response. This is indeed true. The concept of normal stiffness k_n is somewhat equivalent to the uniaxial drained bulk modulus or the drained vertical incompressibility K_v per unit length [13]. It is related to the storage properties of the porous medium by

$$k_n = \frac{K_v}{L} = \frac{\alpha \rho g}{S_s \gamma L}, \quad (46)$$

where α here is Biot-Willis coefficient, S_s is the uniaxial specific storage, and γ is the loading efficiency. The concept of effective normal stiffness k_n^{eff} accounts for both rock and fluid compressibility. It can also be related to the storage properties of the porous medium by

$$k_n^{\text{eff}} = \left[\frac{1}{k_n} + H_0 c_f \right]^{-1} = \frac{1}{L} \left[\frac{1}{K_v} + \phi \frac{1}{K_f} \right]^{-1} = \frac{\rho g}{L S_s}. \quad (47)$$

To address this point, we have added a short description of how k_n^{eff} in the model relates to the uniaxial specific storage in a continuum in the main text when defining the concept of effective normal stiffness (line 78), and in the supplementary materials when deriving the pore pressure equation (section 1.2), and when applying our model to a real world-setting (section 7).

Comment 6. Check that all figures in the main text showing critical kappa are done under conditions where the quasi-steady state approximation is justified. Mark regions where it is invalid as such.

Response. All figures in the main text were produced with dimensionless parameters ($0 < c \ll 1$, $0 < rq \ll 1$). Figure 2A is with ($c = 3 \times 10^{-2}$, $rq = 5 \times 10^{-3}$, error $< 13\%$), Figure 3A is with ($c = 3 \times 10^{-2}$, $rq = 5 \times 10^{-3}$, error $< 13\%$), Figure 3B is with ($c = 3 \times 10^{-2}$, $rq = 2 \times 10^{-2}$, error $< 38\%$), and Figure 3C is with ($c = 3 \times 10^{-2}$, $rq = 2 \times 10^{-3}$, error $< 6\%$). To address this point, we have included the values of dimensionless parameters c and rq in the captions of Figs. 2 and 3.

Comment 7. There have been several more recent studies of the Linker-Dieterich effect. One recent one is Kilgore et al., 2012. That study implies a very specific value of α .

Response. Linker and Dieterich (1992) [14] examined the shear strength following a step change in normal stress, and found that the shear strength responds immediately and then is followed by a gradual response. They modeled this effect by the addition of the $\hat{\alpha}\dot{\sigma}/\sigma$ term to the state evolution equation. The value of $\hat{\alpha}$ ranges from 0.2 to 0.5 for rock with $\mu_0 = 0.7$, which means that roughly 30% to 70% of normal stress effect is gradual and the other 70% to 30% is immediate.

Kilgore et al (2012) [15] examined the same phenomena using newer apparatus, and found that the shear strength has no immediate response, only gradual. Both studies agree that as the normal stress is abruptly raised, the state variable abruptly drops. The results of this study can be captured using the same $\hat{\alpha}\dot{\sigma}/\sigma$ term by simply setting $\hat{\alpha} = \mu_0$, so that 100% of the normal stress effect is gradual and 0% is immediate.

We state in the supplementary material (section 1.1) of the original manuscript that “theoretical and laboratory studies for a sudden change in normal stress show that $\hat{\alpha}$ ranges from 0 to μ , but more studies are needed to determine the value of $\hat{\alpha}$ for a gradual change in normal stress”. We use 1 as a value of α ($= \hat{\alpha}/\mu_*$) in all our analytical and simulation examples. We have also added a reference to Kilgore et al. (2012) [15] in the main text (line 78).

Comment 8. Appendix section 1.2. I think there might be errors or typos in the derivation of the pore pressure equation. First I was confused by the definition of m_{in} and m_{out} . I would have thought $\dot{m}_{in} = q$, and \dot{m}_{out} would be the Darcy outflow. The difference of these would then be set equal to the storage term $p dV_f/dt$. If this is right, then (S6) should not have the $-qdt$ term. Also the Darcy flux equation (S7) should have only m_{out} on the left side. Additionally (S7) is also missing a factor of the fault cross-sectional area, unless permeability is really permeability times cross-sectional area. The cross-sectional area is also missing in (S8). Finally, I think density is missing in (S10).

Response. Although we believe there were no errors in the derivation of the pore pressure equation in the original manuscript as ρ was embedded in the dynamic fluid viscosity and volumetric injection rate in (S10) and $A = 1$ in (S7) and (S8), we acknowledge that the way we defined some variables was confusing. In addition, another reviewer suggested to avoid the assumption that the fluid is incompressible compared to the rock matrix. To

clarify the confusion and account for slightly compressible fluid, we have modified the pore pressure equation derivation (section 1.2):

To obtain a physical evolution of effective stress on the frictional surface, we couple it with a poroelastic model of pore pressure and rock deformation. Starting with the principle of mass conservation, we specify the change of mass from fluid diffusion to be Δm_{diff} , mass accumulation due to rock expansion or fluid compressibility $\frac{\partial}{\partial t}(\rho V_f)\Delta t$, and injection source term to be $\tilde{Q}\Delta t$. We assume that both the fluid and rock matrix are compressible [13], and so mass balance leads to

$$\Delta m_{\text{diff}} = \frac{\partial(\rho V_f)}{\partial t}\Delta t - \tilde{Q}\Delta t, \quad (48)$$

where the change in fluid mass due to pressure diffusion can be written using Darcy's law as

$$\Delta m_{\text{diff}} = -\frac{\rho k A (P - P_0)}{\eta L}\Delta t, \quad (49)$$

where η is fluid dynamic viscosity, k is permeability, and L is the pressure diffusion length. The mass accumulation term can be expressed as

$$\frac{\partial}{\partial t}\rho V_f\Delta t = \frac{\partial}{\partial t}\rho(P)HA\Delta t = \frac{\partial}{\partial t}(\rho_0(1 + c_f(P - P_0))(H_0 + W))A\Delta t, \quad (50)$$

where H is the current height of the slider, ρ_0 is the initial fluid density, c_f is fluid compressibility, H_0 is the initial height of the slider, and W is the position of the piston. When fluid is injected into a rock that is free to deform in the direction orthogonal to sliding, the addition of mass induces an increase of volume equivalent to

$$V_f - V_{f,0} = AW, \quad (51)$$

where $V_{f,0}$ is the initial fluid volume. We then derive an expression for rock deformation W from force balance, while using the convention of compression positive,

$$W = \frac{A}{k_n}(\Sigma'_0 - \Sigma + P), \quad (52)$$

where k_n is the normal spring stiffness and Σ'_0 is the initial effective stress. We further approximate the mass accumulation term to

$$\rho \frac{\partial}{\partial t}[AH]\Delta t + AH \frac{\partial P}{\partial t}\Delta t = \rho \frac{A}{k_n} \frac{\partial P}{\partial t}\Delta t + H \frac{\rho_0}{\rho} c_f \frac{\partial P}{\partial t}\Delta t \approx \rho \frac{A}{k_n} \frac{\partial P}{\partial t}\Delta t + H_0 c_f \frac{\partial P}{\partial t}\Delta t. \quad (53)$$

Note that we consider that the total stress Σ is analogous to overburden stress in the earth, and is therefore constant in time. By substituting Eqs. (49)-(52) into Eq. (48), we find that pore pressure satisfies a diffusion

equation that leads to transient behavior at early times and steady-state behavior at late times

$$\dot{P} = \frac{k_n^{\text{eff}} k}{\eta L A} (P_0 - P) + \frac{k_n^{\text{eff}}}{A} Q, \quad (54)$$

where η is fluid dynamic viscosity ($\eta = \nu \rho$), Q is the volumetric injection rate per unit area ($Q = \tilde{Q}/\rho A$), and $k_n^{\text{eff}} = (1/k_n + c_f H_0/A)^{-1}$ is an effective stiffness somewhat equivalent to the uniaxial bulk modulus or the reciprocal of the uniaxial specific storage per diffusion length in a continuum [13]. Since the slider has a unit base area ($A = 1$), the evolution of the pore pressure as a result of fluid injection follows

$$\dot{P} = \frac{k_n^{\text{eff}} k}{\eta L} (P_0 - P) + k_n^{\text{eff}} Q. \quad (55)$$

Comment 9. *The analysis might be simplified if the reference velocity v_* was chosen to be v_0 , without loss of generality.*

Response. Since we used v_* in our characteristic and normalized quantities to allow for the option $v_0 \neq v_*$ in our simulations, we kept them as different parameters in our analysis.

Comment 10. *State the total number of dimensionless parameters in the system (S15)-(S18) and their conceptual meaning. Have the authors studied the role of each parameter? Maybe this is not necessary. But it is very important to state and justify the realistic range of values of the dimensionless parameters, especially to make connection to the real world and real injection schedules. Here it will be important to explain the role of the diffusion length L as well as the normal stiffness k_n (the latter might be connected to storage properties of a continuum).*

Response. We agree with the reviewer that it is important to identify the dimensionless parameters, and the range of expected values. There is a total of nine dimensionless parameters in the system: $\kappa = (k_s d_c)/\tau_c$, $a = \hat{a}/\mu_c$, $b = \hat{b}/\mu_c$, $\alpha = \hat{\alpha}/\mu_c$, $\epsilon = (T/2\pi)/t_c$, $c = t_c/(\eta L/k_n^{\text{eff}}/k)$, $r = t_c k_n^{\text{eff}}$, and $q = Q/p_c$. The parameter κ is the normalized shear stiffness, and a , b , α are normalized frictional parameters. The parameter ϵ is the normalized oscillation period or ratio of inertial to state-evolution timescales, which may range from 10^{-8} to 10^{-6} depending on rupture diameter and shear wave speed. The parameter c is the normalized diffusivity or ratio of the pore-pressure to the state-evolution timescales, which may range from 10^{-4} to 10^1 depending on reservoir permeability, uniaxial bulk modulus, and well-fault distance. The parameter $r q$ is the normalized injection rate, which may range from 10^{-5} to 10^{-1} depending on injection rate and reservoir size.

To address this comment in the updated manuscript, we have added the definition of the dimensionless parameters and the range of expected values both in the supplementary materials (section 2) and in the main text (line 85), and have modified our discussion of

the role of parameters c and rq in main text (Figure 2B) and supplementary material (section 8).

Comment 11. *Figure S1. Adaptive step size, mentioned in caption, is not shown. Also, slip shown here is probably the usual one (time integral of v) since it increases linearly with time for stable sliding (whereas u would be constant).*

Response. We thank the referee for raising this point. We have removed “adaptive step size” from the caption as we do not include it in the figure. For your reference, we have included the adaptive step size plot below. It is also true that the slip shown here is the usual one. It is related to the relative displacement by $\delta = v_0 t - u$.

FIGURE 2. Slip vs. time, and adaptive time step size

Comment 12. *Figure S2. Is the change in κ_{cr} coming from the change in effective stress (i.e., the usual dependence of critical stiffness on effective stress)? Or it is really being caused by the additional \dot{p} term?*

Response. The confusion seems to stem from the absence of a reference κ_{crit} for comparison. The increase in k_{crit} at early times is caused by the rate term \dot{p} and the decrease in k_{crit} at late times is caused by the magnitude term p . To clarify the confusion, we have plotted the initial k_{crit} without injection (black), the magnitude effect with injection and $\alpha = 0$ (green), and the combined rate-magnitude effect with injection and $\alpha = 1$ (blue).

To further clarify this point, we have added the simulation results without fluid injection (Fig. 4) as a reference for the reader to compare it to the simulation results with fluid injection (Fig. 5) in section 4 of the supplementary materials.

FIGURE 3. Analytical vs. numerical estimates of critical stiffness κ_{crit} for a case with normalized diffusivity of $c = 3 \times 10^{-2}$ and normalized injection rate of $r_q = 5 \times 10^{-3}$. Solid line represents the analytical estimate, and open circles represent numerical ones. Blue represents the combined rate and magnitude effect (setting $\alpha = 1$), and green represents the magnitude effect (setting $\alpha = 0$). The black horizontal line represents κ_{crit} prior to fluid injection.

FIGURE 4. Dynamics of the spring-porosliding system without fluid injection, for velocity-weakening friction. Top figure (A) shows time evolution of normal effective stress ($\sigma - p$), velocity ϵv , state variable θ , and magnitude of the inertia term $\epsilon^2 \dot{v}$. Bottom figure (B) shows plots of normalized stress as a function of velocity, time, and slip. Red curves represent the slip phase, blue curves represent the stick phase, and green curves represent steady-sliding.

FIGURE 5. Dynamics of the spring-porosliding system under constant fluid injection rate, for velocity-weakening friction. Top figure (A) shows time evolution of normal effective stress ($\sigma - p$), velocity ϵv , state variable θ , and magnitude of the inertia term $\epsilon \dot{v}$. Bottom figure (B) shows plots of normalized stress as a function of velocity, time, and slip. Red curves represent the slip phase, blue curves represent the stick phase, and green curves represent steady-sliding.

Comment 13. *The authors are encouraged to connect their results with a continuum linear stability analysis (and nonlinear simulations) with rate-and-state friction in a poroelastic medium by Heimissson et al., 2019. There have also been a few papers showing nonlinear simulations with rate-and-state friction and poroelasticity, including injection: Torberntsson et al., 2018; Pampillon et al., 2018; maybe also by Chunfang Meng at MIT?*

Response. We thank the referee for pointing us to these recent references on continuum stability analysis and nonlinear simulations. We have connected our study with the following:

- (line 116) Heimissson et al. (2019) [16] perform continuum linear stability analysis for fault slip with mildly velocity-strengthening friction ($O(10^{-4})$), and show that slip-induced poroelastic pressure change has a destabilising effect in the undrained limit. Similar to our findings, fluid flow has a stabilizing effect as it acts to equilibrate the pressure change. However, this effect stems from the relation between the rate of change in the frictional shear strength and the change in magnitude of effective normal stress through rate-and-state friction for constant normal load ($\hat{\alpha} = 0$) in Heimissson’s work, whereas it mostly stems from the relation between the frictional shear strength and the rate of change in effective normal stress through Linker and Dieterich (1992) [14] rate-and-state friction for variable normal load ($\hat{\alpha} = \mu_*$) in our work.
- (line 42) Torberntsson et al. (2018) [17] and Pampillon et al. (2018) [18] study fault slip triggered by fluid injection and diffusion in a 2D poroelastic continuum. Stability depends on frictional parameters and magnitude of effective stress only, as neither one uses rate-dependent triggering criteria (critical nucleation length in the former and Coulomb failure in the later).
- (line 123) Segall et al. (2010) [19] show that slow slip events (aseismic) are associated with low effective stress (high pore pressure) and low stress drops.

In summary, we thank the referee for the thoughtful comments, which helped improve our manuscript significantly. We hope that our response has properly addressed the points raised in his review.

REFERENCES

- [1] William A Olsson. The effects of normal stress history on rock friction. In *The 29th US Symposium on Rock Mechanics*, pages 111–117. American Rock Mechanics Association, 1988.
- [2] Lee A Segel and Marshall Slemrod. The quasi-steady-state assumption: a case study in perturbation. *SIAM Review*, 31(3):446–477, 1989.
- [3] Chia-Chiao Lin and Lee A Segel. *Mathematics applied to deterministic problems in the natural sciences*. SIAM, 1988.
- [4] Andy Ruina. Slip instability and state variable friction laws. *Journal of Geophysical Research: Solid Earth*, 88(B12):10359–10370, 1983.

- [5] David M Evans. The Denver area earthquakes and the Rocky Mountain Arsenal disposal well. *The Mountain Geologist*, 3(1):23–36, 1966.
- [6] JH Healy, WW Rubey, DT Griggs, and CB Raleigh. The Denver earthquakes. *Science*, 161(3848):1301–1310, 1968.
- [7] ML Blanpied, DA Lockner, and JD Byerlee. Fault stability inferred from granite sliding experiments at hydrothermal conditions. *Geophysical Research Letters*, 18(4):609–612, 1991.
- [8] CH Scholz. The critical slip distance for seismic faulting. *Nature*, 336(6201):761, 1988.
- [9] Chris Marone. Laboratory-derived friction laws and their application to seismic faulting. *Annual Review of Earth and Planetary Sciences*, 26(1):643–696, 1998.
- [10] Paul A Hsieh and John D Bredehoeft. A reservoir analysis of the Denver earthquakes: A case of induced seismicity. *Journal of Geophysical Research: Solid Earth*, 86(B2):903–920, 1981.
- [11] JD Byerlee. The mechanics of stick-slip. *Tectonophysics*, 9(5):475–486, 1970.
- [12] Richard M Iverson. Regulation of landslide motion by dilatancy and pore pressure feedback. *Journal of Geophysical Research: Earth Surface*, 110(F2), 2005.
- [13] Herbert F Wang. *Theory of linear poroelasticity with applications to geomechanics and hydrogeology*. Princeton University Press, 2000.
- [14] MF Linker and James H Dieterich. Effects of variable normal stress on rock friction: Observations and constitutive equations. *Journal of Geophysical Research: Solid Earth*, 97(B4):4923–4940, 1992.
- [15] Brian Kilgore, Julian Lozos, Nick Beeler, and David Oglesby. Laboratory observations of fault strength in response to changes in normal stress. *Journal of Applied Mechanics*, 79(3):031007, 2012.
- [16] Elías R Heimisson, Eric M Dunham, and Martin Almquist. Poroelastic effects destabilize mildly rate-strengthening friction to generate stable slow slip pulses. *Journal of the Mechanics and Physics of Solids*, 130:262–279, 2019.
- [17] Kim Torberntsson, Vidar Stiernström, Ken Mattsson, and Eric M Dunham. A finite difference method for earthquake sequences in poroelastic solids. *Computational Geosciences*, 22(5):1351–1370, 2018.
- [18] Pedro Pampillón, David Santillán, Juan Carlos Mosquera, and Luis Cueto-Felgueroso. Dynamic and quasi-dynamic modeling of injection-induced earthquakes in poroelastic media. *Journal of Geophysical Research: Solid Earth*, 123(7):5730–5759, 2018.
- [19] Paul Segall, Allan M Rubin, Andrew M Bradley, and James R Rice. Dilatant strengthening as a mechanism for slow slip events. *Journal of Geophysical Research: Solid Earth*, 115(B12), 2010.

UNDERSTANDING RATE EFFECTS IN INJECTION-INDUCED EARTHQUAKES.

Amendments in response to the comments from Referee 3.

We thank the referee for his/her evaluation of the manuscript, which has helped improve our manuscript. The referee’s comments are included verbatim in *italics* and our responses follow each point.

Comment 1. The authors of the manuscript analyse then the stability/instability trade off of their resulting very non-linear (and thus extremely sensitive to many parameter perturbations) equation system in terms of the critical stiffness of the system. When the system gets unstable, they claim an earthquake.

Response. The association between earthquakes and spring-slider systems’ instabilities has long been well-recognized. Brace and Byerlee (1966) [1] proposed that stick-slip frictional instability on pre-existing faults is a source of shallow earthquakes, where “slip” is the earthquake and “stick” is the interseismic period of elastic strain accumulation [2, 3]. This seminal *Science* paper led to the development of rate-and-state friction laws [4, 5, 6, 7], and subsequently to the theoretical analyses of spring-slider systems’ instabilities [6, 8, 9, 10, 11].

The classic instability criterion derived in previous studies, for constant pore pressure, states that the slider slips by stick-slip (analogous to the earthquake cycle) if the stiffness of the loading system is less than the critical value and it slips by steady-sliding (analogous to aseismic creep), otherwise. Here we develop a poroelastic spring–slider model as an analog for induced earthquakes, and derive a new instability criterion for variable pore pressure.

Comment 2. I have not found any clear poroelastic coupling formulation. Such one should take into account mutual impacts of stresses and pressures. This is not the case here.

Response. It is unfortunate that the referee missed the description of the poroelastic coupling in our manuscript: it is derived and explained in detail in the supplementary materials (section 1). The system’s dynamics (Eq. 2) is derived from momentum balance of forces acting on the slider, where the frictional evolution (frictional shear stress) is governed by rate-and-state friction for variable effective normal stress (Eq. 3). Pore pressure evolution (Eq. 4) is derived from mass and force balance, and represents a poroelastic model of pore pressure and rock deformation. The addition of fluid from injection expands the spring (analogous to rock expansion), and thus induces pressure increase inside the slider (analogous to pore pressure or pressure inside the rock pores). Frictional and pore pressure

evolution are coupled through the effective normal stress. Therefore, our model accounts for the poroelastic coupling between the shear and effective normal stresses along the fault.

Comment 3. *The assumption of non-compressibility of the fluid seems to be strange in this particular case. Seismic waves have usually velocities up to 5 times higher than the acoustic waves in the water. Thus, the fluid is significantly more compressive than the rock.*

Response. We thank the referee for bringing up this point. To address it fully, we have modified the derivation of the poroelastic coupling to account for both fluid and rock compressibility (section 1.2 of supplementary material):

To obtain a physical evolution of effective stress on the frictional surface, we couple it with a poroelastic model of pore pressure and rock deformation. Starting with the principle of mass conservation, we specify the change of mass from fluid diffusion to be Δm_{diff} , mass accumulation due to rock expansion or fluid compressibility $\frac{\partial}{\partial t}(\rho V_f)\Delta t$, and injection source term to be $\tilde{Q}\Delta t$. We assume that both the fluid and rock matrix are compressible [12], and so mass balance leads to

$$\Delta m_{\text{diff}} = \frac{\partial(\rho V_f)}{\partial t}\Delta t - \tilde{Q}\Delta t, \quad (1)$$

where the change in fluid mass due to pressure diffusion can be written using Darcy's law as

$$\Delta m_{\text{diff}} = -\frac{\rho k A (P - P_0)}{\eta L}\Delta t, \quad (2)$$

where η is fluid dynamic viscosity, k is permeability, and L is the pressure diffusion length. The mass accumulation term can be expressed as

$$\frac{\partial}{\partial t}\rho V_f\Delta t = \frac{\partial}{\partial t}\rho(P)HA\Delta t = \frac{\partial}{\partial t}(\rho_0(1 + c_f(P - P_0))(H_0 + W)A)\Delta t, \quad (3)$$

where H is the current height of the slider, ρ_0 is the initial fluid density, c_f is fluid compressibility, H_0 is the initial height of the slider, and W is the position of the piston. When fluid is injected into a rock that is free to deform in the direction orthogonal to sliding, the addition of mass induces an increase of volume equivalent to

$$V_f - V_{f,0} = AW, \quad (4)$$

where $V_{f,0}$ is the initial fluid volume. We then derive an expression for rock deformation W from force balance, while using the convention of compression positive,

$$W = \frac{A}{k_n}(\Sigma'_0 - \Sigma + P), \quad (5)$$

where k_n is the normal spring stiffness and Σ'_0 is the initial effective stress. We further approximate the mass accumulation term to

$$\rho \frac{\partial}{\partial t} [AH] \Delta t + AH \frac{\partial P}{\partial t} \Delta t = \rho \frac{A}{k_n} \frac{\partial P}{\partial t} \Delta t + H \frac{\rho_0}{\rho} c_f \frac{\partial P}{\partial t} \Delta t \approx \rho \frac{A}{k_n} \frac{\partial P}{\partial t} \Delta t + H_0 c_f \frac{\partial P}{\partial t} \Delta t. \quad (6)$$

Note that we consider that the total stress Σ is analogous to overburden stress in the earth, and is therefore constant in time. By substituting Eqs. (2)-(5) into Eq. (1), we find that pore pressure satisfies a diffusion equation that leads to transient behavior at early times and steady-state behavior at late times

$$\dot{P} = \frac{k_n^{\text{eff}} k}{\eta L A} (P_0 - P) + \frac{k_n^{\text{eff}}}{A} Q, \quad (7)$$

where η is fluid dynamic viscosity ($\eta = \nu \rho$), Q is the volumetric injection rate per unit area ($Q = \tilde{Q}/\rho A$), and $k_n^{\text{eff}} = (1/k_n + c_f H_0/A)^{-1}$ is an effective stiffness somewhat equivalent to the uniaxial bulk modulus or the reciprocal of the uniaxial specific storage per diffusion length in a continuum [12]. Since the slider has a unit base area ($A = 1$), the evolution of the pore pressure as a result of fluid injection follows

$$\dot{P} = \frac{k_n^{\text{eff}} k}{\eta L} (P_0 - P) + k_n^{\text{eff}} Q. \quad (8)$$

Comment 4. *Equations shown in 1-5 are quite unclear. Are they all dimensionless? However it is not really defined. Characteristic normalizing quantities are not defined. The pore pressure and the pressure inside slider are indistinguishable etc..*

Response. We thank the reviewer for pointing out the confusion. Although we document in the original manuscript that Eqs. (1)-(4) are dimensional, Eq. (5) is dimensionless, and we refer to the supplementary materials (section 2) for the derivation of the dimensionless form of the equations and the definition of the characteristic normalizing quantities, we further clarify the confusion. We now use uppercase letters for dimensional variables and keep lowercase letters for dimensionless variables in our governing equations in both the supplementary materials (sections 1 and 2) and the main text (line 69):

The dimensional equations describing the dynamic motion of the poroelastic spring–slider system with an evolving pore pressure take the form (see

supplementary materials for the derivation of the equations [13]):

$$\dot{U} = V_0 - V, \quad (9)$$

$$\dot{V} = \frac{1}{(T/2\pi)^2} \left[U - \frac{1}{k_s} (\mu_* + \hat{a} \ln \frac{V}{V_*} + \Theta) (\Sigma - P) \right], \quad (10)$$

$$\dot{\Theta} = -\frac{V}{d_c} (\Theta + \hat{b} \ln \frac{V}{V_*}) + \hat{\alpha} \frac{\dot{P}}{(\Sigma - P)}, \quad (11)$$

$$\dot{P} = \frac{k_n^{\text{eff}} k}{\eta L} (P_0 - P) + k_n^{\text{eff}} Q. \quad (12)$$

where U is the relative displacement between the load point and the slider, $(\dot{})$ denotes time derivative, V_0 is the loading velocity, V is slip rate, T is the vibration period, k_s is the shear stiffness, V_* is a normalizing slip rate, μ_* is a constant appropriate for steady-state at slip rate V_* , \hat{a} and \hat{b} are experimentally derived parameters relating friction to changes in slip rate and state, respectively, Θ is a state variable describing the sliding surface, Σ is the total stress, P is the pressure inside the slider (pore pressure), d_c is the characteristic slip distance, $\hat{\alpha}$ is a scaling factor ranging from 0 to μ [14, 15], k_n^{eff} is the effective normal stiffness (related to the uniaxial bulk modulus or the reciprocal of the uniaxial specific storage per diffusion length in a continuum), k is the permeability, η is fluid dynamic viscosity, L is the pressure diffusion length, P_0 is the ambient pressure, and Q is the volumetric injection rate per unit area.

Choosing the following characteristic quantities: $u_c = d_c$, $v_c = V_*$, $\mu_c = \mu_*$, $p_c = P_0$, $\tau_c = \mu_* (\Sigma - P_0)$, $\theta_c = \mu_*$, and $t_c = d_c/V_*$, the equations describing the dynamic motion of the system, in dimensionless form, become (see supplementary materials [13]):

$$\dot{u} = v_0 - v, \quad (13)$$

$$\dot{v} = \frac{1}{\epsilon^2} \left[u - \frac{1}{\kappa} (1 + a \ln v + \theta) (\sigma - p) \right], \quad (14)$$

$$\dot{\theta} = -v (\theta + b \ln v) + \alpha \frac{\dot{p}}{(\sigma - p)}, \quad (15)$$

$$\dot{p} = c(p_0 - p) + r q, \quad (16)$$

where $\kappa = (k_s d_c)/\tau_c$, $a = \hat{a}/\mu_c$, $b = \hat{b}/\mu_c$, $\alpha = \hat{\alpha}/\mu_c$, $\epsilon = (T/2\pi)/t_c$, $c = t_c/(\eta L/k_n^{\text{eff}}/k)$, $r = t_c k_n^{\text{eff}}$, and $q = Q/p_c$. The parameter κ is the normalized shear stiffness, and a , b , α are normalized frictional parameters. The parameter ϵ is the normalized oscillation period or ratio of inertial to state-evolution timescales, which may range from 10^{-8} to 10^{-6} depending on rupture diameter and shear wave speed. The parameter c is the normalized diffusivity or ratio of the pore-pressure to the state-evolution

timescales, which may range from 10^{-4} to 10^1 depending on reservoir permeability, uniaxial bulk modulus, and well-fault distance. The parameter rq is the normalized injection rate, which may range from 10^{-5} to 10^{-1} depending on injection rate and reservoir size.

***Comment 5.** It is completely unclear what would justify its high scientific significance, innovative character and relation to the reality.*

Response. Let us break this comment into two parts; the first related to its scientific significance, and the second related to its relation to reality.

In this work, we propose a minimal-ingredients model of earthquake nucleation that reflects rate effects on induced earthquakes. The model is in the spirit of the classic spring-slider model of frictional slip [2], but couples the mechanics of shear motion with the dynamics of pore pressure governed by fluid injection and pressure diffusion. As we document in the manuscript, a slip instability analysis and numerical simulations show that the key contributing factor to triggering earthquakes is the rate of increase in pore pressure—thus challenging current mechanisms used to explain induced earthquakes based on the magnitude of pore pressure alone [16, 17]. This is, in our view, an exciting and potentially transformative result. It provides, for the first time, a first-principles link between the rate of fluid injection and the likelihood of triggering earthquakes, which—as we show in the manuscript—can be used to design operational strategies for seismic hazard mitigation. We establish this link with a mechanistic, zero-dimensional model, which provides the basis to understand the plethora of recent field observations that have remained, heretofore, unexplained. The significant contribution was reflected in the comments from the other two reviewers, who state that our study “represents a timely and important contribution to the state of the art in physics-based forecasting of induced earthquakes”, “very important”, and “nice idealization of the coupled dynamics of pore pressure evolution and fault slip”.

The comment about the connection with reality is well taken, and this was also brought up by another reviewer.

The 1960s Denver earthquakes is a good example of a real-world setting, where it is well-documented that injection of wastewater into the fractured Precambrian granite gneiss underneath the Rocky Mountain Arsenal triggered the earthquakes and where injection rate is directly related to the frequency of earthquakes [18, 19]. The reservoir spans a depth interval from 3.7 to 7 km below the surface. Experimental data on granite at this depth shows velocity weakening behavior ($\hat{b} - \hat{a}$ in the range 0.002 to 0.005, $\mu_0 = 0.7$ to 0.75) [20].

To identify values of dimensionless parameter c that correspond to this setting, we estimate the state evolution timescale t_Θ and the pore pressure evolution timescale t_p :

$$c = \frac{t_\Theta}{t_p}. \quad (17)$$

The state evolution timescale t_Θ is

$$t_\Theta = \frac{d_c}{V_0}. \quad (18)$$

We find that t_Θ ranges from 10 days to 4 months based on field data of the characteristic slip distance and loading rate ($d_c = 10^{-3}$ to 10^{-2} m, $V_0 = 10^{-9}$ m/s) [21, 7].

The pore pressure evolution time scale t_P can be transferred from the poroslider model to field settings,

$$t_P = \underbrace{\frac{\eta L}{k_n^{\text{eff}} k}}_{\text{poroslider}} = \frac{L}{k/\eta} \left[\frac{1}{k_n} + H_0 c_f \right] = \frac{L^2}{k/\eta} \left[\frac{1}{K_v} + \phi \frac{1}{K_f} \right] = \frac{L^2 S_s}{k/\eta \rho g} = \underbrace{\frac{L^2 S}{T}}_{\text{field}}. \quad (19)$$

We find that t_P is approximately 2 years based on a reservoir analysis of the Denver earthquakes, with transmissivity $T = 10^{-5}$ m²/s, storativity $S = 10^{-5}$, and characteristic length scale $L = 8 \times 10^3$ m [22].

In a similar manner, we identify values of dimensionless parameter rq . We translate this quantity from the poroslider model to field settings:

$$rq = \underbrace{\frac{d_c}{V_0} k_n^{\text{eff}} \frac{Q}{P_0}}_{\text{poroslider}} = \frac{d_c}{V_0} k_n^{\text{eff}} \frac{Q_w}{P_0 W B} = \frac{d_c}{V_0} \frac{\left[\frac{1}{K_v} + \phi \frac{1}{K_f} \right]^{-1}}{L} \frac{Q_w}{P_0 W B} = \underbrace{\frac{d_c \rho g}{V_0 L S} \frac{Q_w}{P_0 W}}_{\text{field}}, \quad (20)$$

and evaluate values based on the reservoir analysis and injection data, with reservoir pressure $P_0 = 30$ MPa, reservoir width $W = 3 \times 10^3$ m, and field injection rate $Q_w = 2$ to 9 million gal/mo [18, 22].

Therefore, reasonable estimates of c and rq for this setting would be in the order of 10^{-2} to 10^{-1} and 10^{-3} to 10^{-1} , respectively. Note that both estimates are much smaller than one, and thus meet the QSSA validity conditions.

Having determined the validity of the QSSA analysis to this setting, we now assess whether pressurization rate effects were likely significant during fluid injection leading to the Denver earthquakes. In dimensionless form, the critical stiffness κ_{crit} is given by Eq. (S29). Prior to fluid injection, the pore pressure is constant and the critical stiffness,

$$\kappa_{\text{crit}} = (b - a)(\sigma - p_0), \quad (21)$$

is estimated to be 0.005 ($b - a = 0.003$ to 0.007 , $\sigma - p_0 = 1$). Shortly following the start of fluid injection, the pore pressure increases rapidly and the dimensionless critical stiffness takes the form:

$$\kappa_{\text{crit}} \Big|_{t=0} = (b - a)(\sigma - p_0) + \frac{\alpha}{v_0} \dot{p} \Big|_{t=0} = (b - a)(\sigma - p_0) + \frac{\alpha}{v_0} rq. \quad (22)$$

This results in an increase in critical stiffness at early times in the range of 30% to 3000% ($\alpha = 1$, $v_0 = 1$), thus indicating that the additional weakening effect from fluid pressurization is likely significant in this setting.

To address this point, we have added a new section in the supplementary materials (section 7) and have added a summary in the main text (line 143).

Comment 6. *The fact that the injection rate (thus, pressure rate) are governing the seismicity rate and thus, the probability of strong earthquakes is very well known. Corresponding theoretical considerations for linear and non-linear poroelastic systems clearly supported by real observations can be found e.g., in Shapiro, Dinske, 2009, JGR, v. 114, B9.*

Response. The fundamental understanding of the link between injection rate and seismicity rate is still an open question. Shapiro & Dinske (2009) [23] did link injection rate to Gutenberg-Richter with anomalous high b values through empirical means, whereas we explain the link between injection rate and likelihood of earthquakes through mechanistic, lab-based friction model to gain deeper physical insights.

Comment 7. *Also the claims of the manuscript of their result similarity with some case studies (such a similarity is not really documented in the manuscript) are not valid. For example, the Basel 2006 event of Mw3.2 (Majer et al, 2007, Geothermics, v.36, 185-222) occurred exactly by the type of the injection shown in the sketch 3 C of the manuscript (recommended by the authors as a stabilizing strategy!). Moreover, the injection was stopped because significant events of Mw close to 3 started to happen and not, because the injection was stopped, the events started to happen (as Fig 3 implies). Also in Pohang, the Mw5.5 occurred already 2 months after the injection was stopped. Moreover, it happened after a cycle with a smallest injection rate.*

Response. We agree with the reviewer that our model does not necessarily explain the occurrence of *all* earthquakes upon fluid injection. As we state in the concluding paragraph of the manuscript, we claim that “our model points to the underlying mechanism by which the rate of fluid pressurization, and hence the rate of effective normal stress unloading, may explain several injection-induced seismicity observations [24, 25, 26, 27, 28, 29, 30, 31]. An abrupt or large increase in injection rate tends to intensify the early-time destabilizing effect of the rate of change in pore pressure, whereas a gradual or small increase in injection rate tends to lessen it”.

While the Basel 2006 event may have resulted from a similar injection schedule, it is unclear from reading Majer et al (2007) [32] whether pore pressure at the fault was allowed to stabilize between injection increments to minimize the early-time destabilizing effect of the rate of increase in pore pressure or not. In addition, other underlying mechanisms may explain the occurrence of the Pohang earthquakes two months after stopping injection with a small injection volume (i.e. pore pressure diffusion, fault activation, etc). This remains an open question of research.

In summary, we thank the referee for the comments, which helped improve our manuscript. We hope that our response has properly addressed the points raised in his/her review, and that he/she is now convinced of the novelty and significance of our contribution.

REFERENCES

- [1] WF Brace and JD Byerlee. Stick-slip as a mechanism for earthquakes. *Science*, 153(3739):990–992, 1966.
- [2] JD Byerlee. The mechanics of stick-slip. *Tectonophysics*, 9(5):475–486, 1970.
- [3] Christopher H Scholz. Earthquakes and friction laws. *Nature*, 391(6662):37, 1998.
- [4] James H Dieterich. Modeling of rock friction: 1. Experimental results and constitutive equations. *Journal of Geophysical Research: Solid Earth*, 84(B5):2161–2168, 1979.
- [5] James H Dieterich. Modeling of rock friction: 2. Simulation of preseismic slip. *Journal of Geophysical Research: Solid Earth*, 84(B5):2169–2175, 1979.
- [6] Andy Ruina. Slip instability and state variable friction laws. *Journal of Geophysical Research: Solid Earth*, 88(B12):10359–10370, 1983.
- [7] Chris Marone. Laboratory-derived friction laws and their application to seismic faulting. *Annual Review of Earth and Planetary Sciences*, 26(1):643–696, 1998.
- [8] James R Rice and Simon T Tse. Dynamic motion of a single degree of freedom system following a rate and state dependent friction law. *Journal of Geophysical Research: Solid Earth*, 91(B1):521–530, 1986.
- [9] K Ranjith and JR Rice. Stability of quasi-static slip in a single degree of freedom elastic system with rate and state dependent friction. *Journal of the Mechanics and Physics of Solids*, 47(6):1207–1218, 1999.
- [10] James H Dieterich and MF Linker. Fault stability under conditions of variable normal stress. *Geophysical Research Letters*, 19(16):1691–1694, 1992.
- [11] Paul Segall and James R Rice. Dilatancy, compaction, and slip instability of a fluid-infiltrated fault. *Journal of Geophysical Research: Solid Earth*, 100(B11):22155–22171, 1995.
- [12] Herbert F Wang. *Theory of linear poroelasticity with applications to geomechanics and hydrogeology*. Princeton University Press, 2000.
- [13] See supplementary material.
- [14] MF Linker and James H Dieterich. Effects of variable normal stress on rock friction: Observations and constitutive equations. *Journal of Geophysical Research: Solid Earth*, 97(B4):4923–4940, 1992.
- [15] Brian Kilgore, Julian Lozos, Nick Beeler, and David Oglesby. Laboratory observations of fault strength in response to changes in normal stress. *Journal of Applied Mechanics*, 79(3):031007, 2012.
- [16] William L Ellsworth. Injection-induced earthquakes. *Science*, 341(6142):1225942, 2013.
- [17] Thibault Candela, Brecht Wassing, Jan Ter Heege, and Loes Buijze. How earthquakes are induced. *Science*, 360(6389):598–600, 2018.
- [18] David M Evans. The Denver area earthquakes and the Rocky Mountain Arsenal disposal well. *The Mountain Geologist*, 3(1):23–36, 1966.
- [19] JH Healy, WW Rubey, DT Griggs, and CB Raleigh. The Denver earthquakes. *Science*, 161(3848):1301–1310, 1968.

- [20] ML Blanpied, DA Lockner, and JD Byerlee. Fault stability inferred from granite sliding experiments at hydrothermal conditions. *Geophysical Research Letters*, 18(4):609–612, 1991.
- [21] CH Scholz. The critical slip distance for seismic faulting. *Nature*, 336(6201):761, 1988.
- [22] Paul A Hsieh and John D Bredehoeft. A reservoir analysis of the Denver earthquakes: A case of induced seismicity. *Journal of Geophysical Research: Solid Earth*, 86(B2):903–920, 1981.
- [23] SA Shapiro and Carsten Dinske. Scaling of seismicity induced by nonlinear fluid-rock interaction. *Journal of Geophysical Research: Solid Earth*, 114(B9), 2009.
- [24] Cliff Frohlich. Two-year survey comparing earthquake activity and injection-well locations in the Barnett Shale, Texas. *Proceedings of the National Academy of Sciences*, 109(35):13934–13938, 2012.
- [25] Matthew Weingarten, Shemin Ge, Jonathan W Godt, Barbara A Bekins, and Justin L Rubinstein. High-rate injection is associated with the increase in US mid-continent seismicity. *Science*, 348(6241):1336–1340, 2015.
- [26] JH Healy, WW Rubey, DT Griggs, and CB Raleigh. The Denver earthquakes. *Science*, 161(3848):1301–1310, 1968.
- [27] Luigi Improta, Luisa Valoroso, Davide Piccinini, and Claudio Chiarabba. A detailed analysis of wastewater-induced seismicity in the Val d’Agri oil field (Italy). *Geophysical Research Letters*, 42(8):2682–2690, 2015.
- [28] Cornelius Langenbruch and Mark D Zoback. How will induced seismicity in Oklahoma respond to decreased saltwater injection rates? *Science Advances*, 2(11):e1601542, 2016.
- [29] Nicolas Cuenot, Catherine Dorbath, and Louis Dorbath. Analysis of the microseismicity induced by fluid injections at the EGS site of Soultz-sous-Forêts (Alsace, France): implications for the characterization of the geothermal reservoir properties. *Pure and Applied Geophysics*, 165(5):797–828, 2008.
- [30] Won-Young Kim. Induced seismicity associated with fluid injection into a deep well in Youngstown, Ohio. *Journal of Geophysical Research: Solid Earth*, 118(7):3506–3518, 2013.
- [31] Lanlan Tang, Zhou Lu, Miao Zhang, Li Sun, and Lianxing Wen. Seismicity induced by simultaneous abrupt changes of injection rate and well pressure in Hutubi gas field. *Journal of Geophysical Research: Solid Earth*, 123(7):5929–5944, 2018.
- [32] Ernest L Majer, Roy Baria, Mitch Stark, Stephen Oates, Julian Bommer, Bill Smith, and Hiroshi Asanuma. Induced seismicity associated with enhanced geothermal systems. *Geothermics*, 36(3):185–222, 2007.

Reviewers' comments:

Reviewer #1 (Remarks to the Author):

The revisions have adequately addressed my review comments.

Reviewer #2 (Remarks to the Author):

The authors have addressed my comments, and I recommend publication. I believe the authors have identified an unexpected but potentially important effect, specifically how pressurization (depressurization) acting over long time scales can decrease (increase) stability of frictional sliding. It is striking that small pressurization rates can apparently lead to substantial changes in stability.

I do have one suggestion, which should be considered optional. Figure S3, which validates the quasi-steady state stability analysis results, is done for parameter values for which the critical stiffness only changes by about 20% from its nominal value. This is in contrast to the claimed change in critical stiffness for the Denver earthquakes example, which the authors estimate to be 30% to 3000%. It would be far more convincing for readers if the authors could validate the quasi-steady state analysis for a case when the critical stiffness change is very large. I do have some worry that the 3000% change is pushing the applicability of the theory outside its range of validity. Numerical simulations demonstrating that the theory is relevant for these extreme cases would be most valuable.

I still remain concerned about directly applying the spring-slider stability analysis to real-world injection scenarios. Changes in "stiffness" from pressurization would correspond to changes in nucleation length. I suspect induced seismicity, especially likelihood of hazard-scale events, is less about nucleation (which is the focus of this study, and appears to be influenced by pressurization) than about favorable stress conditions being established over sufficiently large sections of a fault. However, I think it's up to the authors to decide how far to push the applicability of this analysis.

Congratulations on this important discovery and its clear presentation.

Eric Dunham

Reviewer #3 (Remarks to the Author):

The authors delivered a very detailed responses and cleaned up the mathematical formulations. Thus, in this aspect my comments have been completely satisfied. Thank you!

The manuscript is an nice analysis of a system of ordinary differential equations describing a very schematic non-linear modeling set for a behavior of an assumed friction law. The authors analyze the stability of this non-linear system and formulate corresponding conditions.

The authors motivate application of their modeling by a set of references to paper containing case studies of induced earthquakes.

Their example with the Denver earthquake is very schematic. Exactly the same huge range of dimensionless parameters c and r_q can be applied to any granite - gneiss configuration of a corresponding depth. Many injections have been performed around the world in such environments (e.g., Basel, Soultz, Cooper Basin). However, the earthquake of Denver was a unique one. Thus, this illustration is not really convincing.

As I commented in my first review, the main conclusions of the

manuscript are not new, or at least they are not compared with previously published quite similar ones.

It is still unclear for me why this very schematic model is better describing reality than well known results on the controlling seismicity rate by the injection rate (for example, see equations 27 - 36 of Shapiro & Dinske 2009 - I see in these equations, describing non-linear diffusion, nothing what is related to the b-value neither anything what would be more empiric than an empiric friction law being a basis for the manuscript under consideration; moreover, the conclusion of the manuscript on lines 164-166 is quite well in agreement with this set of equations)

Finally, as I mentioned in my previous review, the recommended by the authors the injection scheme shown on Fig. 3C is exactly one which led to the Basel earthquake (MI 3.2, less than the Denver one but quite dramatic by its consequences). The Pohang Mw 5.5 one occurred 2 months after the injection stop. All these cannot be explained by the scheme of this manuscript. However, all these fact have not found any comment in the manuscript.

UNDERSTANDING RATE EFFECTS IN INJECTION-INDUCED EARTHQUAKES.

Amendments in response to the comments from Referee 2.

We thank the referee for a positive re-evaluation of the manuscript. He states that “the authors have identified an unexpected but potentially important effect, especially how pressurization (depressurization) acting over long time scales can decrease (increase) stability of frictional sliding” and that “it is striking that small pressurization rate can apparently lead to substantial changes in stability”. He concludes by stating “congratulations on this important discovery and its clear presentation”. In what follows, we respond to each of the referee’s remaining concerns in turn. The referee’s comments are included verbatim in *italics* and our responses follow each point.

Comment 1. I do have one suggestion, which should be considered optional. Figure S3, which validates the quasi-steady state stability analysis results, is done for parameter values for which the critical stiffness only changes by about 20% from its nominal value. This is in contrast to the claimed changes in critical stiffness for the Denver earthquakes example, which the authors estimate to be 30% to 3000%. It would be far more convincing for readers if the authors could validate the quasi-steady state analysis for a case when the critical stiffness change is very large. I do have some worry that the 3000% change is pushing the applicability of the theory outside its range of validity. Numerical simulations demonstrating that the theory is relevant for these extreme cases would be most valuable.

Response. We thank the referee for raising this point. The 3000% change might indeed be too high for the QSSA analysis to be accurate, but we are unable to calculate the maximum error associated with these conditions using numerical simulations (the integration algorithm `ode15s` in MATLAB runs into convergence issues). However, we are able to infer the maximum error from the error plot (Fig. S4). The maximum error would be more than 50% for this case ($c = 10^{-2}$, $rq = 10^{-1}$). Thus, to be conservative, we now state the logarithmic average of the change in critical stiffness (300%) instead of the range (30% to 3000%). The analytical estimate of critical stiffness for this case (Fig. 1) has a maximum error around 30%.

To address this point, we have modified the application to the Denver earthquakes (section 7) of SM. Red indicates deleted text and blue indicates added text.

Therefore, reasonable estimates of c and rq for this setting would be in the order of 10^{-2} to 10^{-1} and 10^{-3} to 10^{-1} , respectively. Note that both estimates are much smaller than one, and thus meet the QSSA validity conditions (Fig. 1).

Having determined the validity of the QSSA analysis to this setting, we now assess whether pressurization rate effects were likely significant during fluid injection leading to the Denver earthquakes. In dimensionless form, the critical stiffness κ_{crit} is given by Eq. (S29). Prior to fluid injection, the pore pressure is constant and the critical stiffness,

$$\kappa_{\text{crit}} = (b - a)(\sigma - p_0), \quad (1)$$

is estimated to be 0.005 ($b - a = 0.003$ to 0.007 , $\sigma - p_0 = 1$). Shortly following the start of fluid injection, the pore pressure increases rapidly and the dimensionless critical stiffness takes the form:

$$\kappa_{\text{crit}} \Big|_{t=0} = (b - a)(\sigma - p_0) + \frac{\alpha}{v_0} \dot{p} \Big|_{t=0} = (b - a)(\sigma - p_0) + \frac{\alpha}{v_0} r q. \quad (2)$$

This results in an increase in critical stiffness at early times ~~in the range of 30% to 3000% ($\alpha = 1, v_0 = 1$)~~ of around 300% ($\alpha = 1, v_0 = 1, b - a = 0.005, r q = 10^{-2}$), thus indicating that the additional weakening effect from fluid pressurization is likely significant in this setting.

We have also modified the application paragraph in main text (line 150).

To bridge the gap between the analysis of the idealized spring-poroslider model and the real world, we extend our instability criterion from dimensionless to dimensional form, and identify values of dimensionless parameters c and $r q$ that correspond to real-world settings. The 1960s Denver earthquakes is a good example of a real-world setting, where it is well-documented that injection of wastewater into the fractured Precambrian granite gneiss underneath the Rocky Mountain Arsenal triggered the earthquakes and where injection rate is directly related to the frequency of earthquakes [1, 2]. We find that reasonable estimates of c and $r q$ for this setting are in the order of 10^{-2} to 10^{-1} and 10^{-3} to 10^{-1} , respectively. ~~Both estimates are much smaller than one, and thus meet the QSSA validity conditions. To assess whether pressurization rate effects are significant during fluid injection leading to the Denver earthquakes, we evaluate the contribution from the pressurization rate term to the critical stiffness in Eq. (9). We estimate an increase in critical stiffness at early times in the range of 30% to 3000%, thus indicating that the additional weakening effect from fluid pressurization is likely significant in this setting.~~ We then assess the effect of fluid pressurization by evaluating its contribution to the critical stiffness in Eq. (9). We find that a reasonable estimate for the increase in critical stiffness at early times is around 300%, which indicates that the weakening effect from fluid pressurization is likely significant in this setting (see supplementary materials for more details on the application to the Denver earthquakes [3]).

FIGURE 1. Analytical vs. numerical estimates of critical stiffness κ_{crit} for a case with normalized diffusivity of $c = 1 \times 10^{-2}$ and normalized injection rate of $rq = 1 \times 10^{-2}$. Solid line represents the analytical estimate, and open circles represent numerical ones. Blue represents the combined rate and magnitude effect (setting $\alpha = 1$), and green represents the magnitude effect (setting $\alpha = 0$). The black horizontal line represents κ_{crit} prior to fluid injection.

***Comment 2.** I still remain concerned about directly applying the spring-slider stability analysis to real-world injection scenarios. Changes in "stiffness" from pressurization would correspond to changes in nucleation length. I suspect induced seismicity, especially likelihood of hazard-scale events, is less about nucleation (which is the focus of this study, and appears to be influenced by pressurization) than about favorable stress conditions being established over sufficiently large sections of a fault. However, I think it's up to the authors to decide how far to push the applicability of this analysis.*

Response. We thank the referee for raising this point. We agree that having favorable stress conditions being established over sufficiently large sections of a fault is an important condition for induced earthquakes, but we believe that pressurization rate effect on the nucleation length is also important. When earthquakes nucleate on a fault with velocity-weakening friction, in general, aseismic creep may begin in sections of favorable stress conditions. The aseismically creeping segment then slowly grows in size until it reaches the

nucleation length, and then it breaks out rapidly into a seismic wave-producing rupture. A significant increase in critical stiffness, or equivalently decrease in nucleation length, from fluid pressurization may further facilitate or exacerbate this breakout—potentially increasing the likelihood of triggering earthquakes.

To address this point, we have modified the paragraph in the main text around line 143. Red indicates deleted text and blue indicates added text.

Our findings, if they are applicable to natural faults, hold interesting and important implications for induced seismicity. The poroelastic spring-slider may be viewed as a simple model of a fault segment in contact with a reservoir, steady-sliding as an analog of aseismic creep, and stick-slip as a seismic wave-producing rupture cycle [4, 5]. *The spring stiffness scales inversely with the size of the fault segment* [6]. Within this view, our findings may be generalized to indicate that a slowly creeping fault segment is destabilized and ~~generates an earthquake if the condition for instability in Eq. (9) is met~~ nucleates an earthquake if its size exceeds a critical value (nucleation length), inversely proportional to the critical stiffness in Eq. (9).

We have also added a short paragraph (line 162):

When earthquakes nucleate on a fault with velocity-weakening friction, in general, aseismic creep may begin in sections of favorable stress conditions. The aseismically creeping segment then slowly grows in size until it reaches the nucleation length, and then it breaks out rapidly into a seismic wave-producing rupture. A significant increase in critical stiffness, or equivalently decrease in nucleation length, from fluid pressurization may further facilitate or exacerbate this breakout—potentially increasing the likelihood of triggering earthquakes.

In summary, we thank the referee for the thoughtful comments, and we hope that our response has properly addressed the points raised in his review.

REFERENCES

- [1] David M Evans. The Denver area earthquakes and the Rocky Mountain Arsenal disposal well. *The Mountain Geologist*, 3(1):23–36, 1966.
- [2] JH Healy, WW Rubey, DT Griggs, and CB Raleigh. The Denver earthquakes. *Science*, 161(3848):1301–1310, 1968.
- [3] See supplementary material.
- [4] WF Brace and JD Byerlee. Stick-slip as a mechanism for earthquakes. *Science*, 153(3739):990–992, 1966.
- [5] James R Rice and Simon T Tse. Dynamic motion of a single degree of freedom system following a rate and state dependent friction law. *Journal of Geophysical Research: Solid Earth*, 91(B1):521–530, 1986.
- [6] Christopher H Scholz. Earthquakes and friction laws. *Nature*, 391(6662):37, 1998.

UNDERSTANDING RATE EFFECTS IN INJECTION-INDUCED EARTHQUAKES.

Amendments in response to the comments from Referee 3.

We thank the referee for a thoughtful re-evaluation of the manuscript. In what follows, we respond to each of the referee’s remaining concerns in turn. The referee’s comments are included verbatim in *italics* and our responses follow each point.

Comment 1. Their example with the Denver earthquake is very schematic. Exactly the same huge range of dimensionless parameters c and rq can be applied to any granite - gneiss configuration of a corresponding depth. Many injections have been performed around the world in such environments (e.g., Basel, Soultz, Cooper Basin). However, the earthquake of Denver was a unique one. Thus, this illustration is not really convincing.

Response. The 1960s Denver earthquakes is a well-documented case, where that injection of wastewater into the fractured Precambrian granite-gneiss underneath the Rocky Mountain Arsenal triggered the earthquakes and where injection rate is directly related to the frequency of earthquakes [1, 2]. It is also well-studied and published, where injection, reservoir, and fault data are all publicly available [3]. It is a good demonstration of the significance of the additional weakening effect of fluid pressurization rate on frictional sliding compared to the original velocity weakening effect in a real-world setting.

As we document in the original manuscript (section 7 of SM), depth is only used to estimate the original velocity-weakening effect $b - a$. Dimensionless parameters c and rq are the ones needed to estimate the additional weakening effect from fluid pressurization. The ranges of these parameters depend not only on the rock type, but also on the reservoir transmissivity, storativity, pressure, dimensions, characteristic diffusion length scale, and injection rate—making our application fairly well constrained.

It is possible that the pressurization rate effect is significant in other settings. An analysis of the Basel earthquake, for instance, indicates that the injection duration after each rate increase is not long enough for pore pressure to stabilize at the fault (see response to comment 3) [4, 5]. In addition, an analysis of the Soultz earthquake indicates that seismicity strongly depends on the variations of the injection rate [6]. Thus, these examples do not appear to contradict our findings. We prefer, however, to restrict our application here to the well-documented case of the Denver earthquakes. A detailed analysis of other historic earthquakes can be the subject of a follow-up study.

Comment 2. As I commented in my first review, the main conclusions of the manuscript are not new, or at least they are not compared with previously published quite similar ones. It is still unclear for me why this very schematic model is better describing reality

than well known results on the controlling seismicity rate by the injection rate (for example, see equations 27 - 36 of Shapiro and Dinske 2009 -see in these equations, describing nonlinear diffusion, nothing what is related to the b -value neither anything what would be more empiric than an empiric friction law being a basis for the manuscript under consideration; moreover, the conclusion of the manuscript on lines 164-166 is quite well in agreement with this set of equations)

Response. We thank the referee for pointing the paper by Shapiro and Dinske (2009) [7], which explains the spatial growth of earthquake magnitude distribution at large-scale—an important aspect. However, we believe that our approach and conclusions are altogether new, and distinct from [7]. In particular,

- We develop a mechanistic, lab-based model of frictional sliding, whereas [7] develop a nonlinear diffusion, Gutenberg-Richter statistics-based model of magnitude distributions.
- We derive a new criterion for earthquake nucleation that differentiates between seismic and aseismic slip, unlike the Coulomb failure criterion used in [7].
- Our criterion shows that earthquake nucleation depends on the *rate of increase* in pore pressure rather than its magnitude, whereas the criterion used in [7] depends on the *magnitude of increase* only (see paragraph 32 in [7]). In particular, they state “this is a probability of a critical pressure to be smaller than the maximum pore pressure at the point r achieved in the time interval from the injection start till the time t .”
- We make the distinction between cumulative volume of injected fluid and injection rate (see Fig. 3 in main text of original manuscript), whereas [7] indicate that seismicity is proportional to the cumulative volume and injection rate in a similar manner (see paragraphs 24, 37, 40, 49).
- Our conclusions that “fluid injection at constant rate acts in the direction of triggering seismic rupture at early times followed by aseismic creep at late times” and that “for the same cumulative volume of injected fluid, an abrupt high-rate injection protocol is likely to increase the seismic risk whereas a gradual step-up protocol is likely to decrease it” are also new.
- The novelty of our results was reflected in the comments from the other two reviewers (self-identified, Matthew Weingarten and Eric Dunham), who state that we have “identified an unexpected but potentially important effect, specifically how pressurization acting over long time scales can decrease stability of frictional sliding”, and that it “represents a timely and important contribution to the state of the art in physics-based forecasting of induced earthquakes”.

We elaborate on some of the referee’s points in more detail below.

In the paper [7], the authors consider the large-scale spatiotemporal effects of nonlinear diffusion on the probability of a given magnitude earthquake using Gutenberg-Richter statistics. The goal is to explain the rate of spatial growth of triggered earthquakes. Gutenberg-Richter statistics is used under the assumption that induced seismicity is a

consequence of a power-law-type distribution of preexisting cracks, where the probability of an event depends on its magnitude and the b-value (a regional seismicity constant). Earthquake triggering along the preexisting cracks follows Coulomb failure criterion, which depends on a critical value of pore pressure magnitude. The authors assume that tectonic load and phenomena related to the rate-and-state friction leading to recharging critical cracks are much slower than the diffusional process of pore pressure relaxation. They conclude that earthquake magnitude distribution is proportional to the cumulative volume of injected fluid, injection duration, and rate.

In our paper, we take *an entirely different approach*. We consider the small-scale temporal effects of pore pressure evolution on the stability of a fault segment using a mechanistic model of frictional sliding. The goal is to investigate the physical origins of earthquake triggering. Frictional sliding is described by the laboratory-derived rate-and-state constitutive laws, which trace down the occurrence of earthquakes to the dynamics of friction in terms of fault properties, shear loading, effective normal stress, sliding rate, and the state of the sliding surface (or age of asperity contacts) [8, 9, 10, 11]. Even though these frictional laws are empirically formulated to describe laboratory observations, they have proven to be capable of reproducing a wide range of observed seismic and aseismic fault behaviors ranging from preseismic slip and earthquake nucleation to coseismic rupture and earthquake afterslip [12, 13, 14, 15, 14].

We derive a *new criterion for earthquake nucleation* from the analysis and simulation of steady frictional sliding with an evolving pore pressure. Unlike the Coulomb failure criterion used in [7], our criterion addresses the dynamics of the rupture and whether a fault slips seismically or aseismically. Characterizing fault slip mode is important to mitigate the seismic risk associated with subsurface operations, as it has been observed that an increase in pore pressure magnitude leads to seismic slip in some sites [16, 17, 18, 19, 20, 21] and aseismic slip in others [22, 23, 24, 25, 26, 27]. Our criterion shows that, when pore pressure diffusion is slow compared to the rate-and-state frictional evolution, earthquake nucleation depends strongly on the rate of increase in pore pressure. Consequently, fluid injection at constant rate acts in the direction of triggering seismic rupture at early times, if the criterion for instability is met, followed by aseismic creep at late times. This result has not been stated by [7] or others.

While the model by [7] links induced earthquakes to the cumulative volume of injected fluid, injection duration and rate proportionally, our model makes the distinction between these quantities. It implies that, for the same cumulative volume of injected fluid, an abrupt high-rate injection protocol is likely to increase the seismic risk whereas a gradual step-up protocol is likely to decrease it. We make this claim under the condition that the injection increment is small (see Fig. 2 in main text, section 8 in SM, and Eq. (S30) in SM of the original manuscript) and that the injection duration is long enough to allow pore pressure at the fault to stabilize between stages (see lines 177-178 in main text and section 9 in SM of the original manuscript). This result is also new and has not been stated by [7] or others.

To compare and contrast our work with [7], we have added a new paragraph in the main text (line 45). Red indicates deleted text and blue indicates added text.

Seismicity-rate models generally consider the large-scale spatiotemporal effects of nonlinear diffusion on the probability of a given magnitude earthquake using Gutenberg-Richter statistics [7], but do not address the dynamics of the rupture and, in particular, whether a fault slips seismically or aseismically. Characterizing fault slip mode is essential to mitigate the seismic risk associated with subsurface operations, as it has been observed that an increase in pore pressure magnitude leads to seismic slip in some sites [16, 17, 18, 19, 20, 21] and aseismic slip in others [22, 23, 24, 25, 26, 27]. Here we develop a poroelastic model of induced earthquake nucleation in the manner of spring–sliders [28, 10, 29, 14, 30] based on rate-and-state friction [8, 10], and we study the effect of injection rate on stick-slip frictional behavior—the mechanism for seismic slip [31].

We have also modified the paragraph around lines 164-166 of the original manuscript (line 175 of the new manuscript).

To study the influence of reservoir properties on the critical injection rate, above which earthquakes are induced, we model the occurrence of earthquakes as a function of dimensionless injection rate $rq = d_c k_n^{\text{eff}} Q / (p_c V_*)$ and normalized diffusivity $c = d_c k_n^{\text{eff}} k / (\eta L V_*)$ for velocity-weakening conditions, $b - a > 0$. We find that the dimensionless critical injection rate is directly proportional to the diffusivity c in the high-diffusivity limit, and independent of it in the low-diffusivity limit (Fig. 2B). These findings suggest that reservoir regions with low hydraulic diffusivity and high stiffness are more prone to induced seismicity than regions with high hydraulic diffusivity and low normal stiffness—a result that ~~would not have been obvious to anticipate~~ qualitatively agrees with the triggering front concept [7] (see supplementary materials for more details on the phase diagram of injection-induced seismicity [32]).

***Comment 3.** Finally, as I mentioned in my previous review, the recommended by the authors the injection scheme shown on Fig. 3C is exactly one which led to the Basel earthquake (Ml 3.2, less than the Denver one but quite dramatic by its consequences). The Pohang Mw 5.5 one occurred 2 months after the injection stop. All these cannot be explained by the scheme of this manuscript. However, all these fact have not found any comment in the manuscript.*

Response. As we mentioned in our previous response to this comment, the Basel earthquake does *not* contradict our recommended injection scheme in Fig. 3C. From reading Majer et al. (2007) [16] and Haring et al. (2008) [4], it seems clear that the injection duration was not long enough for pore pressure at the fault (not the well!) to stabilize

between injection increments. According to the analysis in our work, this pressure stabilization is needed to minimize the early-time destabilizing effect of the rate of increase in pore pressure on frictional sliding. To demonstrate this, we estimate that the pore pressure evolution time scale t_p from reservoir data ($k = 4 \times 10^{-15} \text{ m}^2$, $\phi = 0.15$, $\eta = 0.0005 \text{ Pa} \cdot \text{s}$, $c_r = 4.4 \times 10^{-4} \text{ MPa}^{-1}$, $c_f = 4.4 \times 10^{-4} \text{ MPa}^{-1}$, $L = 300 \text{ m}$) [4, 5].

$$t_p = \frac{\eta L}{k_n^{\text{eff}} k} = \frac{L^2}{k/\eta} \left[c_r + \phi c_f \right] \approx 65 \text{ days.} \quad (1)$$

That is, it takes approximately *two months* for pore pressure to diffuse from the injection well to the nearest segment of the fault. In contrast, the duration of each injection stage at Basel was approximately *one day*, which makes pore pressure stabilization at the fault—a key recommendation—highly unlikely.

To address this point, we have added a reference to the Basel earthquake [16] (line 188).

To further understand how injection rate may be used to minimize or mitigate the seismic hazard, we simulate three different injection scenarios, and examine the stability of each. Figure 3 demonstrates how injecting the same volume of fluid can have very different seismic potential depending on the injection profile. We observe that injecting at constant rate in scenario (A) causes the critical stiffness to increase at early times, potentially triggering earthquakes, and decrease at late times, potentially resulting in the cessation of earthquakes. In addition, we observe a dramatic drop in critical stiffness upon stopping injection followed by recovery to the value prior to injection. Scenario (B) shows that a higher injection rate yields higher critical stiffness, implying an increased risk of seismicity for this higher injection rate. Scenario (C), where the injection rate ramps up in stages, seems to be most stable because the maximum critical stiffness is lower than its value in both (A) and (B). *If the duration of each stage is not sufficiently long for pressure to stabilize at the fault, the ramp-up injection strategy does not counteract the destabilizing effect of the rate of pore pressure increase at each injection rate increment. (In the Basel enhanced geothermal site, for example, the duration of injection stages was 1 day [4], while the time for pressure to stabilize at the fault is longer than 1 month.)* This suggests that a gradual increase in injection rate, where pore pressure is allowed to stabilize between injection stages, may be the safest injection strategy.

As we have also mentioned in our previous response, the Pohang earthquake occurrence and timing could have resulted from a complex interplay of many mechanisms (e.g., fluid pressurization rate at the fault, stress changes from aseismic slip, spatial growth of pore pressure diffusion, fault reactivation, etc). We do not claim that our model explains the occurrence of *all* earthquakes upon fluid injection. Understanding the mechanisms behind induced earthquakes is still an open question of research that is strongly needed to improve current mitigation strategies [33].

To emphasize this point, we have modified the concluding paragraph of the main text (line 198).

In summary, our model points to the underlying mechanism by which the rate of fluid pressurization, and hence the rate of effective normal stress unloading, may explain several injection-induced seismicity observations [34, 35, 17, 36, 37, 6, 38, 39]. An abrupt or large increase in injection rate tends to intensify the early-time destabilizing effect of the rate of change in pore pressure on the nucleation length, whereas a gradual or small increase in injection rate tends to lessen it. Our findings, as a whole, suggest injection strategies to mitigate the seismic risk associated with a wide range of subsurface operations, from wastewater injection to geologic CO₂ sequestration. Of course, a complex interplay of different mechanisms such as heterogeneous fault stresses, stress changes from aseismic slip, spatial growth of pore pressure diffusion, and static and dynamic stress transfer often play a role in the occurrence and, in particular, the timing of an earthquake [21]. This emphasizes the need to continue to develop and test new models for the forecast and control of induced seismicity [33].

In summary, we thank the referee for the comments, which have helped improve our manuscript. We hope that our response has properly addressed the points raised in his/her review and, in particular, reemphasized the novelty in our approach, analysis, and results.

REFERENCES

- [1] David M Evans. The Denver area earthquakes and the Rocky Mountain Arsenal disposal well. *The Mountain Geologist*, 3(1):23–36, 1966.
- [2] JH Healy, WW Rubey, DT Griggs, and CB Raleigh. The Denver earthquakes. *Science*, 161(3848):1301–1310, 1968.
- [3] Paul A Hsieh and John D Bredehoeft. A reservoir analysis of the Denver earthquakes: A case of induced seismicity. *Journal of Geophysical Research: Solid Earth*, 86(B2):903–920, 1981.
- [4] Markus O Häring, Ulrich Schanz, Florentin Ladner, and Ben C Dyer. Characterisation of the basel 1 enhanced geothermal system. *Geothermics*, 37(5):469–495, 2008.
- [5] Jack Norbeck and Roland Horne. Injection-triggered seismicity: An investigation of porothermoelastic effects using a rate-and-state earthquake model. In *Proceedings of the 40th Workshop on Geothermal Reservoir Engineering, Stanford*, 2015.
- [6] Nicolas Cuenot, Catherine Dorbath, and Louis Dorbath. Analysis of the microseismicity induced by fluid injections at the EGS site of Soultz-sous-Forêts (Alsace, France): implications for the characterization of the geothermal reservoir properties. *Pure and Applied Geophysics*, 165(5):797–828, 2008.
- [7] SA Shapiro and Carsten Dinske. Scaling of seismicity induced by nonlinear fluid-rock interaction. *Journal of Geophysical Research: Solid Earth*, 114(B9), 2009.

- [8] James H Dieterich. Modeling of rock friction: 1. Experimental results and constitutive equations. *Journal of Geophysical Research: Solid Earth*, 84(B5):2161–2168, 1979.
- [9] James H Dieterich. Modeling of rock friction: 2. Simulation of preseismic slip. *Journal of Geophysical Research: Solid Earth*, 84(B5):2169–2175, 1979.
- [10] Andy Ruina. Slip instability and state variable friction laws. *Journal of Geophysical Research: Solid Earth*, 88(B12):10359–10370, 1983.
- [11] MF Linker and James H Dieterich. Effects of variable normal stress on rock friction: Observations and constitutive equations. *Journal of Geophysical Research: Solid Earth*, 97(B4):4923–4940, 1992.
- [12] Christopher H Scholz. Earthquakes and friction laws. *Nature*, 391(6662):37–42, 1998.
- [13] Sylvain Barbot, Nadia Lapusta, and Jean-Philippe Avouac. Under the hood of the earthquake machine: Toward predictive modeling of the seismic cycle. *Science*, 336(6082):707–710, 2012.
- [14] Paul Segall and James R Rice. Dilatancy, compaction, and slip instability of a fluid-infiltrated fault. *Journal of Geophysical Research: Solid Earth*, 100(B11):22155–22171, 1995.
- [15] Chris Marone. Laboratory-derived friction laws and their application to seismic faulting. *Annual Review of Earth and Planetary Sciences*, 26(1):643–696, 1998.
- [16] Ernest L Majer, Roy Baria, Mitch Stark, Stephen Oates, Julian Bommer, Bill Smith, and Hiroshi Asanuma. Induced seismicity associated with enhanced geothermal systems. *Geothermics*, 36(3):185–222, 2007.
- [17] JH Healy, WW Rubey, DT Griggs, and CB Raleigh. The Denver earthquakes. *Science*, 161(3848):1301–1310, 1968.
- [18] CB Raleigh, JH Healy, and JD Bredehoeft. An experiment in earthquake control at Rangely, Colorado. *Science*, 191(4233):1230–1237, 1976.
- [19] Katie M Keranen, Matthew Weingarten, Geoffrey A Abers, Barbara A Bekins, and Shemin Ge. Sharp increase in central Oklahoma seismicity since 2008 induced by massive wastewater injection. *Science*, 345(6195):448–451, 2014.
- [20] William L Yeck, GP Hayes, Daniel E McNamara, Justin L Rubinstein, William D Barnhart, PS Earle, and Harley M Benz. Oklahoma experiences largest earthquake during ongoing regional wastewater injection hazard mitigation efforts. *Geophysical Research Letters*, 44(2):711–717, 2017.
- [21] F Grigoli, S Cesca, AP Rinaldi, A Manconi, JA López-Comino, JF Clinton, R Westaway, C Cauzzi, T Dahm, and S Wiemer. The November 2017 Mw 5.5 Pohang earthquake: A possible case of induced seismicity in South Korea. *Science*, 360(6392):1003–1006, 2018.
- [22] Scott D Davis and Wayne D Pennington. Induced seismic deformation in the Cogdell oil field of West Texas. *Bulletin of the Seismological Society of America*, 79(5):1477–1495, 1989.
- [23] Yves Guglielmi, Frédéric Cappa, Jean-Philippe Avouac, Pierre Henry, and Derek Elsworth. Seismicity triggered by fluid injection–induced aseismic slip. *Science*, 348(6240):1224–1226, 2015.

- [24] FH Cornet, J Helm, H Poitrenaud, and A Etchecopar. Seismic and aseismic slips induced by large-scale fluid injections. In *Seismicity associated with mines, reservoirs and fluid injections*, pages 563–583. Springer, 1997.
- [25] Seid Bourouis and Pascal Bernard. Evidence for coupled seismic and aseismic fault slip during water injection in the geothermal site of soultz (france), and implications for seismogenic transients. *Geophysical Journal International*, 169(2):723–732, 2007.
- [26] Mark D Zoback, Arjun Kohli, Indrajit Das, Mark William McClure, et al. The importance of slow slip on faults during hydraulic fracturing stimulation of shale gas reservoirs. In *SPE Americas Unconventional Resources Conference*. Society of Petroleum Engineers, 2012.
- [27] Shengji Wei, Jean-Philippe Avouac, Kenneth W Hudnut, Andrea Donnellan, Jay W Parker, Robert W Graves, Don Helmlberger, Eric Fielding, Zhen Liu, Frederic Cappa, et al. The 2012 brawley swarm triggered by injection-induced aseismic slip. *Earth and Planetary Science Letters*, 422:115–125, 2015.
- [28] JD Byerlee. The mechanics of stick-slip. *Tectonophysics*, 9(5):475–486, 1970.
- [29] James R Rice and Simon T Tse. Dynamic motion of a single degree of freedom system following a rate and state dependent friction law. *Journal of Geophysical Research: Solid Earth*, 91(B1):521–530, 1986.
- [30] Richard M Iverson. Regulation of landslide motion by dilatancy and pore pressure feedback. *Journal of Geophysical Research: Earth Surface*, 110(F2), 2005.
- [31] WF Brace and JD Byerlee. Stick-slip as a mechanism for earthquakes. *Science*, 153(3739):990–992, 1966.
- [32] See supplementary material.
- [33] Kang-Kun Lee, William L Ellsworth, Domenico Giardini, John Townend, Shemin Ge, Toshihiko Shimamoto, In-Wook Yeo, Tae-Seob Kang, Junkee Rhie, Dong-Hoon Sheen, et al. Managing injection-induced seismic risks. *Science*, 364(6442):730–732, 2019.
- [34] Cliff Frohlich. Two-year survey comparing earthquake activity and injection-well locations in the Barnett Shale, Texas. *Proceedings of the National Academy of Sciences*, 109(35):13934–13938, 2012.
- [35] Matthew Weingarten, Shemin Ge, Jonathan W Godt, Barbara A Bekins, and Justin L Rubinstein. High-rate injection is associated with the increase in US mid-continent seismicity. *Science*, 348(6241):1336–1340, 2015.
- [36] Luigi Improta, Luisa Valoroso, Davide Piccinini, and Claudio Chiarabba. A detailed analysis of wastewater-induced seismicity in the Val d’Agri oil field (Italy). *Geophysical Research Letters*, 42(8):2682–2690, 2015.
- [37] Cornelius Langenbruch and Mark D Zoback. How will induced seismicity in Oklahoma respond to decreased saltwater injection rates? *Science Advances*, 2(11):e1601542, 2016.
- [38] Won-Young Kim. Induced seismicity associated with fluid injection into a deep well in Youngstown, Ohio. *Journal of Geophysical Research: Solid Earth*, 118(7):3506–3518, 2013.
- [39] Lanlan Tang, Zhou Lu, Miao Zhang, Li Sun, and Lianxing Wen. Seismicity induced by simultaneous abrupt changes of injection rate and well pressure in Hutubi gas field.

Journal of Geophysical Research: Solid Earth, 123(7):5929–5944, 2018.

REVIEWERS' COMMENTS:

Reviewer #3 (Remarks to the Author):

I find, the authors did an excellent job for making their paper understandable and well readable. Now, this is really a good work. However, I think that this work is erroneously targeted to explaining induced seismicity. It is very nice that the last revision takes a little bit more into account of the existing understanding and of observations of this phenomenon. However, the obtained results are in a strong disagreement with this phenomenon, which unfortunately, has been chosen to motivate the author's stability analysis of a rate-and state friction-controlled and poro-elastically coupled spring-slider model.

Thus, the publication fate of this manuscript is up to the editor and up to the authors. However, I do want to add a couple of more detailed concluding comments:

- The vibration period T of equation (2) has not been clearly defined. It seems that it is related to one of internal frequencies of the system and then, it should be related to the characteristic diffusion length. However, mutual relation of various characteristic times and their observational physics remains unclear. Moreover, the notation " T " has been used later for defining normalized time and for denoting hydraulic conductivity (supplements).
- The authors state that "In the Basel enhanced geothermal site, for example, the duration of injection stages was 1 day, while the time for pressure to stabilize at the fault is longer than 1 month." What is the basis of this statement (in its last part)? Which fault do they mean? There were several involved ones, if not hundreds.
- Corresponding to the main illustrative result of the manuscript, Figure 3, the most hazardous time periods of induced seismicity are nearly immediately after or rather close to starting moments or moments of jumping increases of injections. The less hazardous moments are after injection terminations. In contrast, common observations of induced seismicity (including Denver, Basel and nearly all other referred to in the manuscript case studies) show strongest events at the end or even after the injection periods.

UNDERSTANDING RATE EFFECTS IN INJECTION-INDUCED EARTHQUAKES.

Response to the comments from Referee 3.

We thank the referee for the re-evaluation of the manuscript. The reviewer states that we “did an excellent job for making the paper understandable and well readable,” and that “now, this is really a good work.” In what follows, we respond to each of the referee’s remaining comments in turn. The referee’s comments are included verbatim in *italics* and our responses follow each point.

***Comment 1.** The vibration period T of equation (2) has not been clearly defined. It seems that it is related to one of internal frequencies of the system and then, it should be related to the characteristic diffusion length. However, mutual relation of various characteristic times and their observational physics remains unclear. Moreover, the notation “ T ” has been used later for defining normalized time and for denoting hydraulic conductivity (supplements).*

Response. Following the dimensional equations describing the poroslider dynamics (now Eqs. (2)–(5) of Methods), we state that T is the vibration period. This is a classic quantity in the spring–slider literature, defined as $T \equiv 2\pi\sqrt{m/k_s}$, where m is the mass of the slider block, and k_s is the spring stiffness [e.g., 1]. This definition was already given, explicitly, in the Supplementary Information, below Eq. (5) there. The symbol T is never used again in the main manuscript. We do use it again as an interim variable in Supplementary Note 6 and, to avoid confusion, we have replaced it by \hat{t} . The use of T as symbol for transmissibility (not hydraulic conductivity) in Supplementary Note 7 is classical in hydrogeology, and should not in any way lead to any confusion with the vibration period of the spring–slider system.

***Comment 2.** The authors state that “In the Basel enhanced geothermal site, for example, the duration of injection stages was 1 day, while the time for pressure to stabilize at the fault is longer than 1 month.” What is the basis of this statement (in its last part)? Which fault do they mean? There were several involved ones, if not hundreds.*

Response. We addressed this question in detail in our response to Comment 3 of the earlier round of reviews, with reference to the relevant papers that show the faults involved in the seismicity, and choosing the *nearest* fault that was shown in Majer et al. (2007) [2] and Haring et al. (2008) [3]. The disparity between the pressure diffusion time

(~ 2 months) and the duration of the injection stages (~ 1 day) makes it clear that pressure had not stabilized at even the nearest segment of the faults involved in the seismicity.

Comment 3. *Corresponding to the main illustrative result of the manuscript, Figure 3, the most hazardous time periods of induced seismicity are nearly immediately after or rather close to starting moments or moments of jumping increases of injections. The less hazardous moments are after injection terminations. In contrast, common observations of induced seismicity (including Denver, Basel and nearly all other referred to in the manuscript case studies) show strongest events at the end or even after the injection periods.*

Response. While there are cases in which the timing of the earthquake occurs after injection (e.g., the Pohang earthquake), *many* observations, which we refer to in our paper, point to the increased occurrence of earthquakes at high injection rates [4, 5, 6, 7, 8, 9, 10, 11]. This is also observed in laboratory experiments, including just-published observations [12] that post-date our manuscript. It is these observations that we explain by means of a simple mechanistic proslider model with rate-state friction. As we reflect in the closing paragraph of the manuscript, “a complex interplay of different mechanisms such as heterogeneous fault stresses, stress changes from aseismic slip, spatial growth of pore pressure diffusion, and static and dynamic stress transfer often play a role in the occurrence and, in particular, the timing of an earthquake [13].”

REFERENCES

- [1] James R Rice and Simon T Tse. Dynamic motion of a single degree of freedom system following a rate and state dependent friction law. *Journal of Geophysical Research: Solid Earth*, 91(B1):521–530, 1986.
- [2] Ernest L Majer, Roy Baria, Mitch Stark, Stephen Oates, Julian Bommer, Bill Smith, and Hiroshi Asanuma. Induced seismicity associated with enhanced geothermal systems. *Geothermics*, 36(3):185–222, 2007.
- [3] Markus O Häring, Ulrich Schanz, Florentin Ladner, and Ben C Dyer. Characterisation of the basel 1 enhanced geothermal system. *Geothermics*, 37(5):469–495, 2008.
- [4] Cliff Frohlich. Two-year survey comparing earthquake activity and injection-well locations in the Barnett Shale, Texas. *Proceedings of the National Academy of Sciences*, 109(35):13934–13938, 2012.
- [5] Matthew Weingarten, Shemin Ge, Jonathan W Godt, Barbara A Bekins, and Justin L Rubinstein. High-rate injection is associated with the increase in US mid-continent seismicity. *Science*, 348(6241):1336–1340, 2015.
- [6] JH Healy, WW Rubey, DT Griggs, and CB Raleigh. The Denver earthquakes. *Science*, 161(3848):1301–1310, 1968.
- [7] Luigi Improta, Luisa Valoroso, Davide Piccinini, and Claudio Chiarabba. A detailed analysis of wastewater-induced seismicity in the Val d’Agri oil field (Italy). *Geophysical Research Letters*, 42(8):2682–2690, 2015.

- [8] Cornelius Langenbruch and Mark D Zoback. How will induced seismicity in Oklahoma respond to decreased saltwater injection rates? *Science Advances*, 2(11):e1601542, 2016.
- [9] Nicolas Cuenot, Catherine Dorbath, and Louis Dorbath. Analysis of the microseismicity induced by fluid injections at the EGS site of Soultz-sous-Forêts (Alsace, France): implications for the characterization of the geothermal reservoir properties. *Pure and Applied Geophysics*, 165(5):797–828, 2008.
- [10] Won-Young Kim. Induced seismicity associated with fluid injection into a deep well in Youngstown, Ohio. *Journal of Geophysical Research: Solid Earth*, 118(7):3506–3518, 2013.
- [11] Lanlan Tang, Zhou Lu, Miao Zhang, Li Sun, and Lianxing Wen. Seismicity induced by simultaneous abrupt changes of injection rate and well pressure in Hutubi gas field. *Journal of Geophysical Research: Solid Earth*, 123(7):5929–5944, 2018.
- [12] L. Wang, G. Kwiatek, E. Rybacki, A. Bonnelye, M. Bohnhoff, and G. Dresen. Laboratory study on fluid-induced fault slip behavior: The role of fluid pressurization rate. *Geophysical Research Letters*, 47:e2019GL086627, 2020.
- [13] F Grigoli, S Cesca, AP Rinaldi, A Manconi, JA López-Comino, JF Clinton, R Westaway, C Cauzzi, T Dahm, and S Wiemer. The November 2017 Mw 5.5 Pohang earthquake: A possible case of induced seismicity in South Korea. *Science*, 360(6392):1003–1006, 2018.